# Scaling Proprioceptive-Visual Learning with Heterogeneous Pre-trained Transformers

**Lirui Wang**[1]    **Xinlei Chen**[2]    **Jialiang Zhao**[1]    **Kaiming He**[1]
[1]MIT CSAIL    [2]Meta, FAIR

https://liruiw.github.io/hpt

## Abstract

One of the roadblocks for training generalist robotic models today is heterogeneity. Previous robot learning methods often collect data to train with one specific embodiment for one task, which is expensive and prone to overfitting. This work studies the problem of learning policy representations through *heterogeneous pre-training* on robot data across different embodiments and tasks at scale. We propose Heterogeneous Pre-trained Transformers (HPT), which pre-train a large, shareable trunk of a policy neural network to learn a task and embodiment agnostic shared representation. This general architecture aligns the specific proprioception and vision inputs from distinct embodiments to a short sequence of tokens and then processes such tokens to map to control robots for different tasks. Leveraging the recent large-scale multi-embodiment real-world robotic datasets as well as simulation, deployed robots, and human video datasets, we investigate pre-training policies across heterogeneity. We conduct experiments to investigate the scaling behaviors of training objectives, to the extent of 52 datasets. HPTs outperform several baselines and enhance the fine-tuned policy performance by over 20% on unseen tasks in multiple simulator benchmarks and real-world settings.

## 1 Introduction

Building robotic policies today is hard: it often requires collecting *specific* data for each robot, task, and environment, and the learned policies do not generalize beyond these specific settings. A historical lesson that has revolutionized machine learning is that pre-training [34, 27, 29] on large-scale, high-quality, and diverse data can bring *general* models that usually outperform specific models. Recent progress in open-source large-scale data collection [14, 76] has made this path possible, but the heterogeneity (such as varying robot hardware and different environments) present in large-scale robotic data has posed a significant challenge. A central question for the field now is how to leverage the heterogeneous robot data to pre-train robotic foundation models [3].

Foundation models from natural language processing [60, 58] and computer vision [37] have shown a paradigm to achieve general-purpose task-agnostic models through pre-training on massive amounts and diversity of data. In addition to the benefits from more data, training with diverse tasks also enforces the representation to be more generalized. These foundation models can achieve high task success rates for various tasks, are more robust to outliers, and are flexible for adapting to new tasks. These approaches map input signals from distinct domains and tasks into a high-dimensional representation space, and exhibit consistent scaling behaviors [34, 29]. After that, minimal fine-tuning is required to transfer the representation for downstream tasks to achieve good performance.

38th Conference on Neural Information Processing Systems (NeurIPS 2024).

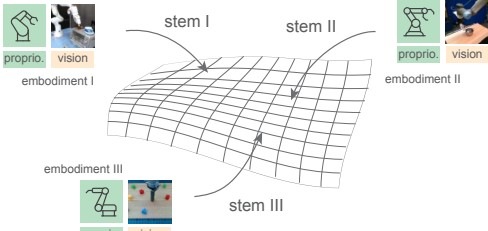

Figure 1: The **Heterogeneous Pre-training** concept. It maps different embodiments, each with its own *proprioception* and *vision* sensors, onto a *shared* latent space by embodiment-specific tokenizers ("stems"). This *aligns* the heterogeneous data from different embodiments into a joint representation space. This allows us to train a shared Transformer trunk on the union of all heterogeneous datasets. The pre-trained Transformer can be transferred to a new embodiment, with a small, new tokenizer learned at transferring time.

The heterogeneity in robotics presents a distinct challenge: different robots are physically different embodiments[1] of hardware acting in different environments. Each embodiment can have a distinct *proprioception*, including different degrees of freedom, end-effectors, motion controllers, and workspace configurations built for a specific application. Another common heterogeneity in robotics is *vision* heterogeneity. Robots are often equipped with different camera sensors mounted at different places (e.g. wrist and/or third-person) and the visual appearance of each robot varies dramatically due to environments and tasks. Both proprioception and vision information are crucial for complex, contact-rich, long-horizon behaviors in robotics. Poor learning of such information can lead to overfitting behaviors such as repeating motions for a particular scene and task or even trajectory.

In this work, we propose to address this issue by aligning the proprioception and vision information from different embodiments to a shared "language" of policies through *heterogenous pre-training* (Figure 1). With such a shared representation, a new embodiment only requires minimal data and training to "translate" its specific setup to the shared "languages". In other words, we want to pre-train task-agnostic and embodiment-agnostic foundational models that can map raw sensor signals from individual embodiments into a shared latent space. Previous works have made significant progress in pre-training only the vision part of the policy on human videos [49, 54, 35, 62] and pre-training the full policy [6, 14, 56] with a unified model and dataset format (e.g. using languages [5]). Additionally, they assume no proprioception in pre-training and add it post hoc in transfer learning.

We introduce Heterogeneous Pre-trained Transformers (HPT), a family of architecture designed to scalably learn from data across heterogeneous embodiments. HPT modularizes a general policy network architecture (Figure 2) and pre-trains the *policy representation* of a latent transformer with supervised learning. Inspired by learning from multimodal data [1, 74, 20, 31], we use *embodiment-specific* tokenizers, dubbed "stem", to align various sensor inputs such as camera views and proprioception inputs. The "trunk" is *shared* and pre-trained across datasets and is transferred when adapting to new embodiments and tasks that are unknown during the pre-training times. Moreover, we use task-specific action decoders, dubbed "head", to produce the action outputs. Crucially, after "tokenizing each embodiment", HPT operates on a shared space of a short sequence of *latent tokens*. This hierarchy is motivated by how humans handle feedback loops between specific motor responses and perceived stimuli at the level of the spinal cord's neural circuitry [69].

We extensively investigated the scaling behaviors and various designs of policy pre-training to the extent of more than 50 individual data sources (2 times more than [56]) and model size of over 1 billion parameters. Analogous to the scaling laws [27, 29], we found that to some extent, HPT scales with the dataset quantity and diversity as well as the model and training compute.

In addition, heterogeneity can occur in different embodiment domains, such as real robot hardware, simulation domains, and human videos. We incorporate many available embodied datasets in different embodiments such as real robots [14, 76, 39], simulation [82, 90, 50, 21, 86, 81] and internet human videos [15] in the pre-training process and demonstrate the generality of our framework including embodiments beyond expensive real-world on-robot teleoperations.

Through transfer learning experiments across multiple simulation benchmarks [90, 50, 82] and real-world dexterous tasks, we compare with several baselines and the from-scratch counterparts. Overall, based on the pre-training objectives, HPT can scale with the model, data, compute, and the heterogeneity of the robotic datasets across real robots, simulations, and human videos. These pre-

---

[1]Embodiment can be defined differently according to the context of robotics and AI. In this work, we consider robots equipped with a distinct set of sensors and actuators with the associated observation and action space to be a unique embodiment.

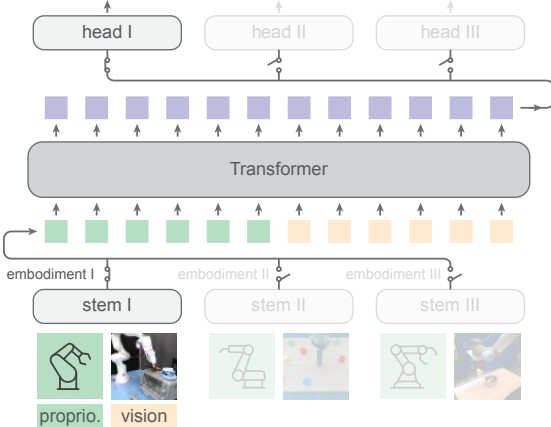

Figure 2: **HPT architecture.** HPT is modularized into stems, trunk, and heads. The stem, consisting of a proprioception tokenizer and a vision tokenizer, maps the vision and proprioception observations of different embodiments to a fixed number (e.g. 16) of tokens. The *shared* trunk, which is a Transformer, maps the concatenated tokens into shared representations. The head then maps the processed tokens to actions in different downstream tasks. For a specific embodiment, one stem/head pair is activated (denoted by the switch). The trunk is shared and pre-trained on action-labeled data with supervised learning and then transferred to new embodiments. This procedure scales up to 52 datasets and 1B parameters.

training procedures and models can simplify building reliable robotic policies for new embodiments and new tasks in terms of data requirements and generalized performance. As an attempt to scale heterogeneous pre-training, our code and weights are open-sourced, and we hope that HPT can shed some light on learning robot representations from heterogeneous embodiments and tasks.

## 2   Related Works

**Pre-training and Transfer Learning.**   Pre-training [2], through direct supervision [38] and/or self-supervision [57, 25, 12, 22, 10], has been shown to learn representation useful for unseen downstream tasks in computer vision [7, 37] and natural language [58], and their intersections [60]. The representation learned from ImagetNet [16] or web-scale data [38, 60, 18] shows robustness to distribution shifts and can be transferred to new tasks.

The recent surge of foundation models [3] scales these representation learning methods [27, 29] by applying task-agnostic objectives to multitask data. Moreover, recent works [46, 42, 45] show that small projection layers can be used to align the pre-trained feature spaces of the foundation models. Different from other fields, robotics has less data quantity and diversity but much more heterogeneity.

**Alignment.** Recent works such as Flamingo [1], Perceiver [31], and ImageBind [20] proposed ways to combine representations from *multimodal data* such as image, language, and audio by aligning these different modalities to the same latent space in the pursuit of representation learning. Our architecture design is also motivated by methods such as LLaVA[45] in the multimodal learning community. Very recently, GPT-4o [58], Gemini [77], MM1 [52], X-VILA [89], and Chameleon [75] demonstrated the capabilities of heterogeneous pre-training a universal transformer from and for multiple modalities. The idea of alignment, across modalities and/or embodiments, is important as we scale to use heterogeneous embodiments and reuse data from distinct embodiments.

**Representation Learning in Robotics.**   Representation learning has been explored in the robotic community. Previous works such as R3M [54], VC-1[49], Voltron[35], and SpatialVLM [11] investigate visual representations by training the policy with human videos and robotic data [70]. Recent works [61, 17, 4, 88, 71, 40] also align representations from multiple modalities and data distributions for robotic tasks. After pre-training, transfer learning with the frozen representation and/or finetuning is conducted in the target domains.

**Generalist Policies.** Large-scale policy learning in robotics has leveraged diverse data from real robots [6, 73], human videos [54, 49], and simulation domain [33, 63, 83, 80] separately. There are also works in multi-task learning [65, 66, 85, 23], meta-learning [79, 55, 19], few-shot learning [84], and fleet learning [82]. Recently, RT-X, Octo, OpenVLA [6, 14, 56, 36] train generalist vision-language-action robotic policies on datasets from diverse robotic embodiments.

Compared with these works, HPT handles broader heterogeneity including proprioception and vision, explores scaling behaviors on more heterogeneous domains including real robots, human videos, and simulation data, and is evaluated at a larger scale in simulation benchmarks.

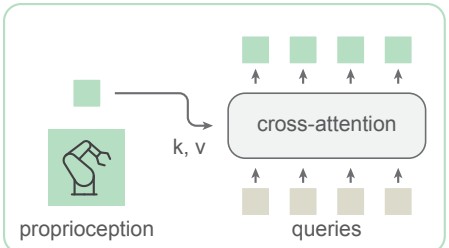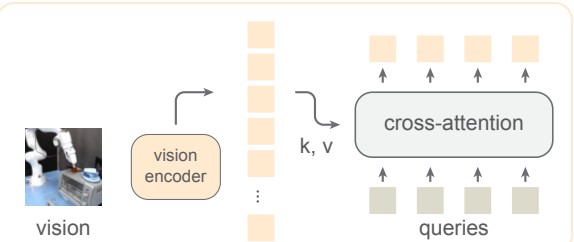

Figure 3: **Stem Architecture in HPT.** In the HPT stem, the proprioceptive tokenizer uses an MLP to map proprioceptive information to a feature which is then attended by 16 learnable tokens. The vision tokenizer uses pre-trained encoders and similarly uses an attention mechanism to map vision features into 16 fixed tokens. The architecture flexibly handles sequences of inputs without increasing the size of tokens.

**Mixture of Experts.** Our architecture design is related to works in conditional computation and MoE [51, 44, 72], where we create one expert for each embodiment, and the router (for the whole network) is determined by the embodiment. This technique has been used to scale language models to a substantial size [32].

## 3    Heterogenoues Pre-trained Transformers (HPT)

In *heterogeneous robot learning* with cross embodiments, the data are generated from different domains such as simulation and real robots, across sensory modalities such as RGB images, language instructions, depth maps, 3D point clouds, and tactile images. Each robot is a unique hardware embodiment with varying degrees of freedom, end-effectors, sensor configurations, controller and action spaces, and application-specific physical setups.

In the following sections, we discuss the HPT network architecture and the training procedure to address the heterogeneity above. We modularize the network architecture (Figure 2) into the embodiment-specific stem, the shared trunk, and the task-specific heads. Intuitively, the stems, shown in Figure 3, are earlier layers of the neural network that align sensory inputs from heterogeneous embodiment and modalities into the shared representation space. The *shared* middle part of the network is called the trunk, which processes the sensory representation into a latent representation that can be used for multiple tasks. Finally, the last part of the network is the head, which maps that latent representation to the action space in individual tasks of interest. The training procedure, dubbed *heterogeneous pre-training*, assigns and aligns specific stem/head pairs based on the sampled embodiment and task data, and still enjoys the benefits of joint training in the shared trunk. This can be thought of as tokenizing each embodiment using neural networks and alleviating the need to unify embodiments into a homogeneous data form in standard training procedures.

### 3.1    Network Architecture

**Stem.** The stem $\theta_{\text{stem}}$ (Figure 3) in HPT is composed of a proprioceptive tokenizer and a vision tokenizer. These tokenizers map heterogeneous inputs from different embodiments to a *fixed number* of tokens with *fixed dimensions*, which enables the trunk to treat them in the same manner despite large heterogeneity, as well as enjoy the scaling and inference benefits on fixed context length. The key idea is to leverage attention [78, 31, 9] to attend a fixed number of learnable tokens to features of the observations. Although we mainly focus on proprioception and vision, handling other kinds of sensor heterogeneity in tactile, 3D, and action inputs can be flexibly extended in stems.

- **Proprioception Tokenizers.** In Figure 3 (left), for embodiment $k$, the proprioceptive tokenizer maps any sequence of robot proprioceptive information with dimension $d_p^k$ (e.g. 7 for end-effector pose) into $N_p$ (e.g. $N_p = 16$) tokens with dimension $d$ with values ranging from 128 to 1024. To achieve this, we first use an MLP to map the proprioceptive input into a feature space with dimension $d$. We then apply sinusoidal position encoding and use attention across the state feature and the learnable tokens, to map into 16 tokens with dimension $d$. Proprioceptive information is critical in robot policy learning, but its usage is often as simple as feature concatenation with a vision encoder [41].

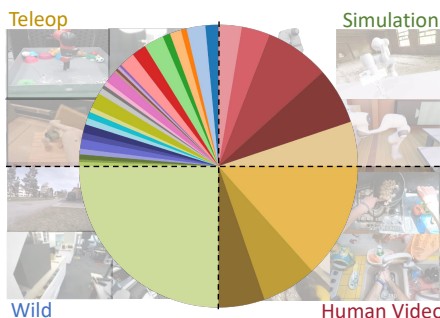

Figure 4: **Dataset Heterogeneity in Robotics.** We show illustrations of dataset mixtures (each color is a distinct embodiment) from different domains including real robot teleop [14], deployed robots [39], simulations, and human videos [15]. See Appendix Section A for dataset mixture details.

|  | # Depth | # Width | # Attn. Heads | # Param. |
|---|---|---|---|---|
| **HPT-Small** | 16 | 128 | 8 | 3.1M |
| **HPT-Base** | 16 | 256 | 8 | 12.6M |
| **HPT-Large** | 16 | 512 | 8 | 50.5M |
| **HPT-XLarge** | 32 | 768 | 16 | 226.8M |
| **HPT-Huge** | 80 | 1024 | 16 | 1.1B |

Table 1: **Network Details of HPT.** The width denotes the latent dimension size of the trunk transformer and the depth denotes the number of blocks. The *default* setup is the HPT-Small model.

|  | # Dataset | # Traj. | # Samples | # Batch Size |
|---|---|---|---|---|
| **Default** | 27 | 16k | 5M | 256 |
| **Scaled** | 52 | 270k | 155M | 2048 |

Table 2: **Dataset Details of Pre-train Settings.** The *default* setup is trained with 27 datasets from RT-X with 16k trajectories (maximum 1000 trajectories each) and *scaled* setup involves more data and compute.

- **Vision Tokenizers.** In Figure 3 (right), the vision tokenizer can map any sequence of camera images (videos of multiple views) with dimension $H \times W \times 3$ into $N_v$ (we use $N_v = 16$ by default) tokens with dimension $d$. To achieve this, we first use *pre-trained frozen feature networks* (e.g. 7 by 7 features from ResNet) and then flatten the features. After that, we again use attention across these features and learnable tokens, to map the vision input into 16 tokens with dimension $d$.

After processing each modality individually in the time sequence order, we concatenate all modality tokens and add additional modality embeddings and sinusoidal positional embeddings. This is used as the input sequence to the trunk that we introduce below. To avoid overfitting, the stem only has a small number of parameters (one MLP and one attention layer).

Related works such as Octo [56] and others [54, 49, 6] mostly focus on pre-training the vision backbone of the policy through masking or self-supervision. They often stack sequences of single-view images along channels [6] for a particular robot or use a large number of tokens (256 in [56]). In contrast, HPT uses stems with pre-trained vision encoders to map arbitrary image sequences to a short sequence of tokens (16). Moreover, rather than *add in* proprioception during transfer in related works, HPT *jointly pre-trains* the vision and proprioception parts, from heterogeneous datasets.

**Trunk.** As the central component for pre-training, the trunk architecture follows a transformer, parametrized by $\theta^{\text{trunk}}$ in the latent space with dimension $d$. The output token sequence length $L$ is the same as the input token sequence length. The output token sequence is simply pooled as the final combined feature for the observation. The trunk is shared across different embodiments and tasks to capture the complex input-output relationships (i.e. the number of trunk parameters is fixed independent of the number of embodiments and tasks).

**Head.** The policy head $\theta_{\text{head}}$ takes the output of the trunk transformer and maps it to the action space $\mathcal{A}$ in each dataset. For each embodiment and task, the policy head can be an arbitrary architecture (e.g. MLP) that takes as input the pooled feature of the trunk and outputs a normalized action trajectory. The policy head is reinitialized for transferring to a new embodiment.

## 3.2 Training Objective

Given a total of $K$ datasets with heterogeneous embodiments sampled from different distributions $\mathcal{D}_1, ..., \mathcal{D}_k, ..., \mathcal{D}_K$, we let $\mathcal{D}_k = \{\tau^{(i)}\}_{1 \leq i \leq M_k}$ denote a set of $M_k$ trajectories in dataset $\mathcal{D}_k$, with $\tau^{(i)} = \{o_t^{(i)}, a_t^{(i)}\}_{1 \leq t \leq T}$ denoting the $i$-th trajectory of maximum length $T$ of observation and action tuples. The objective is to minimize the following loss across datasets

$$\min_{\theta} \sum_{k=1}^{K} \mathcal{L}(\theta_k^{\text{stem}}, \theta^{\text{trunk}}, \theta_k^{\text{head}}; \mathcal{D}_k). \tag{1}$$

$\mathcal{L}$ is behavior cloning loss computed as the Huber loss between the normalized action labels based on dataset statistics and the network's action predictions. $\theta = \bigcup_{k=1}^{K}\{\theta_k^{\text{stem}}, \theta_k^{\text{head}}\} \cup \theta^{\text{trunk}}$ denotes the network parameters comprised of embodiment-specific stem and head $\theta_k^{\text{stem}}, \theta_k^{\text{head}}$ for dataset $k$, and a single set of shared trunk parameters $\theta_{\text{trunk}}$ across all embodiments. This training procedure has two axes of data scaling: the quantity $M_k$ for one dataset $D_k$ and the total number of datasets $K$. In the pre-training stage, only the trunk parameters are updated at every iteration, and the stems and heads for each heterogeneous embodiment and task are updated based on the training batch sampling. See implementation details in Appendix Section A.3.

### 3.3 Transfer Learning

The policy transfer process is similar to aligning the features of the new domain (through pre-trained stem encoders) to the pre-trained embedding space of the trunk [42, 45]. Given a new dataset $\mathcal{D}_{K+1}$ from a new embodiment, the objective can be the same as pre-training or alternatives [13]. We reinitialize the head and stem parameters with embodiment-specific input and output dimensions (such as different proprioception and action dimensions), and freeze the weights of the trunk.

## 4 Experiments on Pre-training

In this section, we aim to answer the following question: Does HPT pre-training have a *scaling behavior* under heterogeneous data across domains?

**Default Setting.** We use 27 robot teleoperation datasets, including a subset of the recently public Open-X Embodiment dataset [14] as the training corpus. By default, we use one camera view of the scene with the pre-trained frozen ResNet18 image encode to compute the vision features. We use proprioception information, such as end-effector poses and joint positions, whenever they are available and provided. We use a maximum of 1000 trajectories from each dataset and a total number of 16k trajectories, and a held-out validation dataset with a maximum 200 trajectories per data source. Furthermore, we use a model with a trunk size of 3.17 million parameters, which is denoted as HPT-Small (Table 1). The training uses a batch size of 256 for 80k iterations, which is around 0.65B tokens in the latent space that feeds into HPTs and around 5B tokens in the vision and proprioception token spaces (horizon-dependent). While we do not align or preprocess action space or observation space [56, 87] other than normalization, data cleanup and filtering would be very helpful.

**Scaled Setting.** We use 200k trajectories with 52 datasets, including simulation (e.g. [50]), deployed robots (e.g. [39]), human videos (e.g. [15]), from distinct embodiments in the training process. This includes many public and accessible robotic datasets. In addition to different tasks in different institutes, these heterogeneous mixtures of datasets (Fig. 4 and Fig. 13) come with multiple views, language inputs, and different observation inputs in different environments.

### 4.1 Protocol

We evaluate the HPT pre-training performance with the *averaged validation loss* (prediction errors on unseen trajectories) at the last iteration of pre-training. These validation datasets are fixed independent of the trajectory counts and models during training. Unless particularly noted, the validation datasets come from the same 27 datasets in the *Default Setting*. Note that it is unrealistic to evaluate the pre-trained models on many real-world robotic environments at scale and there are very few evaluation alternatives to measure large-scale pre-training if we ignore this objective. In fields such as NLP[30, 34], training loss objective (e.g. perplexity) is often used to measure the progress of pre-training. Admittedly, there are several caveats to this metric including the closed-loop performance gap and the task success rate gap. We will address these issues in Section 5 on HPT transfer learning. See Appendix Section A and Section D for more details and discussions.

### 4.2 Scaling Behaviors

**Data Scaling.** In Figure 5 (a), we observe stable and scaling validation losses even on increasingly heterogeneous embodiments. Moreover, we found the compute (e.g. samples seen per training run) and the data amounts needed to scale in tandem [34] to get closer to convergence in the training process. In the red line in Figure 5 (a), we observe better validation losses as we scale up the total number of trajectories, by using a larger model and doubling the batch size every order of magnitude increase in trajectory counts. Strictly increasing data while keeping others bottlenecked (HPT-S

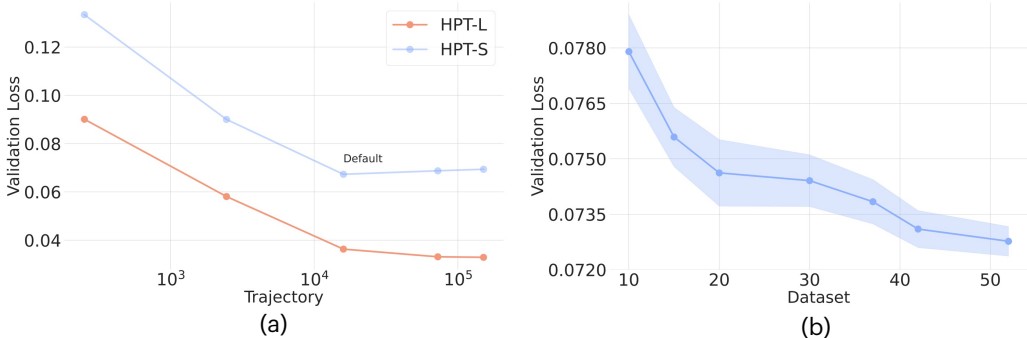

Figure 5: **Data Scaling**. We run scaling HPT experiments along dataset sizes and the number of datasets. Each point is the validation loss of a full training run. (a) We evaluate the losses on 27 datasets with the number of total trajectories ranging from a maximum of 10 trajectories per dataset (270 in total) to a maximum of 100000 trajectories per dataset (170k in total). We compare two model sizes, HPT-S/L, where HPT-L is a bigger model trained with 4 times more tokens than HPT-S. (b) We compute the validation losses for a fixed subset of 10 datasets with a fixed number of epochs (2). We compute mean and standard deviations for 4 runs across model sizes from HPT-S to HPT-XL and across dataset counts from 10 to 52.

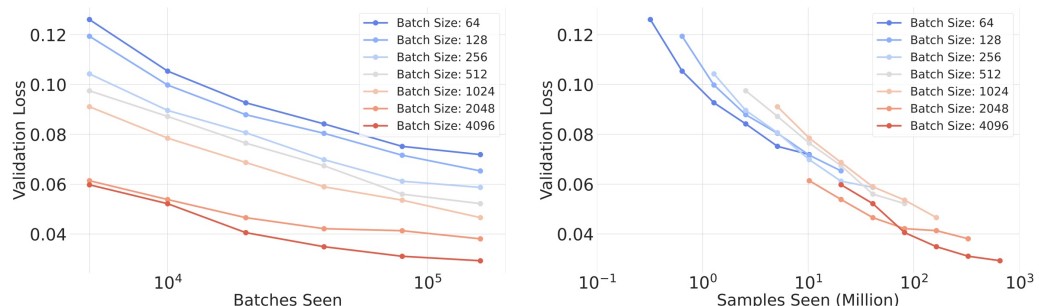

Figure 6: **Epoch Scaling**. We run scaling HPT experiments along the number of total samples. Each point is the validation loss of a full pre-training run. Setting: HPT-S, 27 datasets with a maximum of 1000 trajectories for each dataset. Left) We scale up the number of batch sizes and measure the changes in validation losses. Right) Derived from the left figure, we multiply the batches seen by the number of samples in each batch.

and fixed iterations) might cause an early plateau performance at around 1000 trajectories max per dataset, as shown in the blue line in Figure 5. In Figure 5 (b), we also pre-train on an increasing number of datasets with a fixed number of epochs and evaluate on the fixed subset (first 10 datasets). We hypothesize that training with more embodiments contributes to the generalization of the trunk. These experiments can scale to the extent of 200k trajectories and 52 datasets.

**Model Scaling.** In Figure 7, we fix the number of datasets (27) in RT-X and use a maximum of 1000 trajectories for each dataset. We scale along model size (from 1M to 1B) and gradually increase the batch sizes from 256 to 2048 (doubles every order of model size increase) and use the larger dataset with 170k trajectories. We observe that when we scale to bigger models with larger amounts of compute (red line), the pre-training can achieve low validation losses until it is plateaued. We do not find a significant difference between scaling depth or scaling width.

**Epoch Scaling.** In this experiment, we fix the number of datasets (27) and use a maximum of 1000 trajectories for each dataset. In Figure 6, we observe that increasing batch sizes (Left), which effectively scales training tokens (Right), can generally improve the model performance until convergence. Another observation we have is to use distributed workers to load from as many datasets as possible to aggregate each batch. We hypothesize that the large variance of training on heterogeneous datasets can be reduced by using a large batch size. See Appendix B for more experiment details.

### 4.3   Pre-training on Synthetic Data and Internet Human Videos

We experiment beyond real-world robot teleop data, which is expensive to collect and scale. For the additional datasets, we consider 7 simulation datasets across many popular simulators Drake [82], Mujoco [90, 50], Isaac Sim [21], and PyBullet [86, 81], as well as Sapien [53] and Flex [67], with

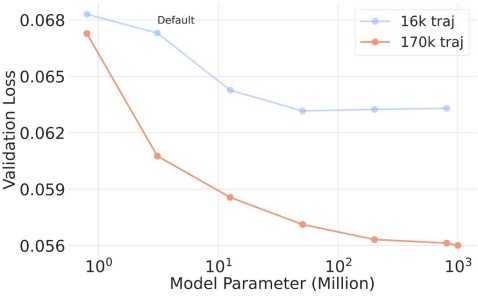

Figure 7: **Model Scaling**. We run scaling HPT experiments along model sizes. Each point is a full training run. Setting: 27 datasets with a maximum of 1000 trajectories for each dataset. We scale along model size (from 1M to 1B) for both the blue and red lines. The red line is trained with increasing data and epochs to reach convergence. Specifically, we gradually increase the batch sizes from 256 to 2048 (doubles every order of model size increase) and use 170k trajectories.

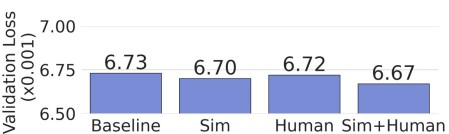

Figure 8: **Joint Pre-training with Simulation and Human Videos.** The baseline denotes the default setting without simulation and human datasets. Setting: We run the experiments with a training corpus of datasets with 1000 trajectories maximum.

image inputs and expert demonstrations. For the human datasets that lack proprioception and action information, we use poses and 2D positions as surrogates for the supervised policy learning objectives. We use in total 300 trajectories from EPIC kitchen [15] and PoCo [83] with a maximum trajectory length 1000. See Appendix Figure 13 and Table 4 for more details on the dataset compositions.

In Figure 8, we use a maximum of 1000 trajectories for each dataset and compare against the baseline of 27 datasets with evaluation on all the pre-trained datasets. We show that pre-training on additional embodiment datasets such as simulation and human video datasets can be possible, despite the large embodiment gaps with real robots. These datasets provide complimentary embodiment data to pure teleop data, and they illustrate how much heterogeneity can be handled in the HPT framework.

## 5 Experiments on Transfer Learning

In the previous section, we evaluate pre-training using the validation losses. In this section, we answer the following question with task success rates in transfer learning: Can the pre-trained HPT model be transferred to new embodiments, tasks, and environments in simulation and the real world?

### 5.1 Transfer to Embodiments in Simulations

**Protocol.** We evaluate the pre-trained representations on robot manipulation simulation benchmarks Meta-world [90], RoboMimic [50], and Fleet-Tools [82]. Each training dataset uses from 20-100 trajectories per task and each testing covers 50 episodes with different initial conditions. The policies use HPT-Small as the pre-trained trunk and reinitialize the stem and head for transferring.

During the evaluation phase, we compare the following models: `No Trunk` uses only the stem and head without the trunk in the middle and trains from scratch as common practice [41]. `From Scratch` trains the entire policy from scratch with the trunk, `Pretrained Frozen` uses and freezes the pretrained trunk during transfer learning and `Pretrained Finetuned` loads the pre-trained HPT-Base trunk and finetunes the whole network end-to-end, and `Pretrained Finetuned (HPT-XL)` uses the same fine-tuning procedure with a pre-trained HPT-XL trunk with a lower pre-training validation loss. To reduce the variance, we conduct independent training runs and evaluations 5 times and average for each model. The inference time during transfer on an RTX 3070 GPU is 47Hz for HPT-base and 19Hz for HPT-XL, while a more recent GPU like A100 can be 3-4 times faster.

**Experiment.** In Figure 10 (a), we test the model on the downstream tasks in closed-loop simulation and observe improved task success rate using the pre-trained models ranging from HPT-B to HPT-XL, although pre-training for the simulation experiments only happens in the real-world embodiments.

In Figure 10 (b), we run HPT on the recently released Simpler [43] Benchmark, which allows for comparing with Octo [56], RT1-X, and RT2-X [14] on a high-fidelity simulation. We focus on three different tasks `Close Drawer`, `Move Near`, and `Pick Coke Can` in the Google EDR embodiment. For each task, we test several different initializations with a total of over 300 episodes for all tasks.

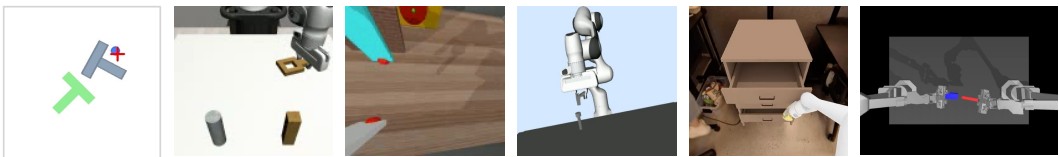

Figure 9: **Simulation Evaluation Tasks.** We evaluate HPT across several simulation benchmarks and show policy rollout visualizations of the experiments. Experiment details can be found in Section 5.1 and A.4.

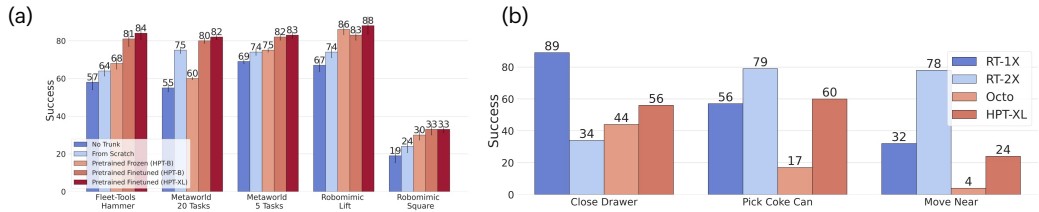

Figure 10: **Success Rates in Simulation Experiments.** (a) We evaluate transfer learning performance of models from HPT-B to HPT-XL on tasks across 4 different simulator benchmarks. (b) We compare with several generalist models in the recent Simpler [43] benchmark with Google GDR embodiment. The pre-trained trunks are trained the Scaled Settings. The success rates are computed over 150 rollouts per approach.

Note that the pre-training corpus of HPT-S does not include [6], and simulation tasks have a focus on language conditioning and do not expose proprioception inputs, which is not suitable for HPT. To address these issues, we finetune HPT on the supervised datasets with around 50 trajectories under the simulation protocol. We use HPT-base as the backbone for this experiment. We use the baseline results from [43]. See Section A.4 for more implementation and experiment details.

## 5.2 Transfer to Embodiments in the Real World

**Protocol.** For the real-world experiments, we evaluate the HPTs on two different embodiments for tasks in pet care and assembly, which are not covered in the pre-training datasets [14]. In particular, for these two robots, we experiment with different observation spaces 1 camera v.s. 2 cameras as well as different action spaces relative pose v.s. absolute pose. For data collection, we experiment with both an Oculus Quest to collect relative pose control as action labels as well as kinesthetic teaching. The episode lengths of real-world teleoperation vary from 50 steps to 150 steps with 10 Hz control frequencies. We experiment with the tasks `Sweep Leftover`, `Fill Water`, `Scoop Food` and `Switch Insertion`, which require 5-20 seconds of interactions with granular or small objects with fine contacts, shown in Figure 11. We collect around 100 demos for each task and evaluate them for 15 trials to measure the average success rate.

**Experiment.** We adopt a similar transfer learning method in the previous section and evaluate the pre-trained HPT representations under real-world evaluation protocols. We train the policy with 20000 iterations with a batch size of 256 and a learning rate of $5e^{-6}$. We defer implementation details to Appendix Section A.5. Quantitatively in Figure 12, we observe pre-trained policies attain a better success rate over the `No-Trunk` and the `From-Scratch` baselines. In particular, the `From-Scratch` baselines in `Fill-Water` use the state-of-the-art diffusion policy architecture to illustrate the flexibility of the pre-trained representations. In Figure 11, qualitatively, we observe better generalization and robustness to varying poses and numbers of granular objects, and varying camera configurations and lighting conditions with pre-trained HPT.

On Table 3, we perform an ablation study for the `Sweep Leftover` task. We also compare with R3M [54], Voltron [35], and VC-1 [49]. We use a finetuned model with the released backbone and weights. We note that these previous works focus on only pre-training the vision encoders of the policies with human videos. Finally, we compared with policies that train from scratch (`From Scratch`) and policies that do not use proprioception during pre-training (`No Prop. Finetuned`) and add in proprioception afterwards. All of our experiments use pre-trained encoders and the trainable parameters (stem and head) can be as few as 2% of the parameters.

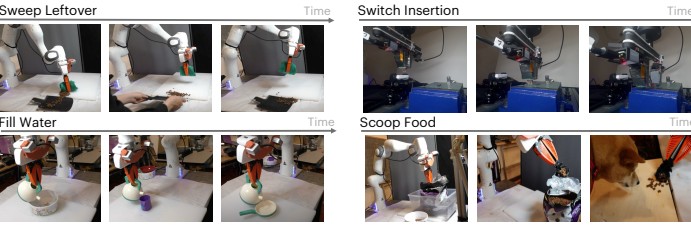

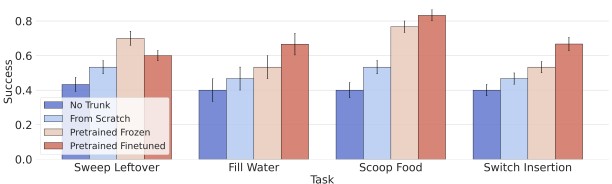

Figure 11: **Real World Qualitative Results**. Pre-trained HPT policies can perform dynamic and long-horizon contact-rich precision tasks in pet care and assembly. The policies show robust and generalized behaviors under scene changes and disturbances.

Figure 12: **Transfer Learning in the Real World.** We evaluate the pre-trained HPTs on four tasks / two embodiments. The average success rate with standard deviations is computed for 45 trials per approach. We use the default pre-training setup with HPT-Base for this experiment. See Section 5.2 for detailed descriptions.

| Method | Success (%) |
|---|---|
| From Scratch No Prop. | 26.7±3.3 |
| From Scratch | 43.3±3.8 |
| R3M [54] | 50.0±3.0 |
| Voltron [35] | 46.7±3.8 |
| VC-1 [49] | 53.3±2.6 |
| No Prop. Finetuned | 63.3±2.6 |
| HPT-B Finetuned | 70.0±3.0 |
| HPT-XL Finetuned | **76.7**±3.3 |

Table 3: **Comparison on the `Sweep Leftover`.** We compare the fine-tuned HPT models with several baselines including vision-only pre-trained models.

## 6    Conclusion

There is room for improvement for many aspects including the dataset curation and pre-training objectives. Specifically, the embodiment splits in our balanced dataset mixture are rather simple. Moreover, careful data filtering to ensure the data quality is under-explored in this work. Also, this work has focused on supervised learning as the pre-training objective and the data size in tokens and training compute sizes in FLOPs only reach a moderate scale of LLM training to ensure full convergence. Although the model architecture and training procedure are modular and independent of embodiment setups, heterogeneous pre-training can converge slowly. For evaluation, both the simulation and real-world evaluation tasks are restricted to short-horizon manipulation tasks with a fixed embodiment, which might limit the benefits of using a higher-capacity model. Furthermore, the learned policies still do not offer very high reliability on the tested tasks (typically below 90%). See Appendix §C for some failure modes.

Given the recent surge of scaled data, robot learning is still limited by its generality because of the heterogeneity, including different embodiments, tasks, and environments where the robots are operated. To handle the heterogeneity common in robotics, we propose HPT, a modular architecture and framework to embrace this heterogeneity through pre-training. We explore and scale HPT with heterogeneous datasets to over 50 available datasets. The learned representation can be transferred and improve performance in both simulation and the real world, and it shows correlations with pre-training performance. The code[2] is open-source for future research. We hope this perspective will inspire future work in handling the *heterogeneous nature* of robotic data for robotic foundation models.

**Acknowledgement.**    We would like to thank Russ Tedrake for discussions and suggestions, Liane Xu for helping with real-world experiments, Tianhong Li for helping with cluster experiments, and Remi Cadene for helping with the LeRobot implementation. We thank MIT Supercloud for providing computing resources to process training data. This work is supported in part by the Amazon Greater Boston Tech Initiative and Amazon PO No. 2D-06310236. Toyota Research Institute provided funds to partially support this work.

---

[2]https://github.com/liruiw/HPT and https://github.com/liruiw/lerobot

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

# A Implementation Details

**Experiment Details.** We conduct pre-training experiments across several orders of magnitudes in computes and data. The number of trajectories and transitions in the dataset is limited by the maximum number of trajectories in each of the constituent datasets. We use maximum episode counts per dataset ranging from 10 trajectories to 100000 trajectories, and the total trajectories range from around 300 trajectories and 6000 transitions to around 300k trajectories and 5 million data points. When training with 80k iterations, the approximate training epochs with fixed batch size 512 range from 200 epochs to 2 epochs. In terms of tokens, our experiment model ranges from 0.5 million to 1 billion parameters, the dataset tokens from all modalities range from approximately 32 million tokens to 5 billion tokens, and the tokens in a batch range from 0.03 million tokens to 2 million tokens (including sequence length). The compute FLOPs range from 0.03GFlops to 31GFlops. See Table 5 for more details of the scale and see Figure 13 for example lists of dataset mixtures. To facilitate future research, we will open-source the data processing scripts.

Different from previous work [56, 87], we use minimal amounts of processing and cleaning of the observation and actions in the raw trajectories. Specifically, the default training setup is to train 80000 iterations with a batch size 256, which is around 0.65B tokens in the latent space that feeds into HPTs and around 5B tokens in the perception token spaces of the raw perception inputs (such as image patches). Due to resource limits, for some bigger datasets such as Droid [76], we did not process the full size.

## A.1 Dataset Details

**Real Robot Teleoperation Dataset.** In total, we use a subset of 42 datasets in the Open-X Embodiment dataset [14], including the recent Droid [76] dataset. These public datasets have high heterogeneity including distinct embodiment, environments to operate on, tasks to solve, etc.

**Simulation Dataset.** For the additional 7 simulation dataset, we use the simulator benchmarks across all popular simulators Drake [82], Mujoco [90, 50], Isaac Sim [21], and PyBullet [81], as well as Sapien [53] and Flex [67], with image inputs and expert demonstrations. These are used as additional training data from the simulation domains.

**Human Video Dataset.** Since the human datasets do not contain proprioception and action information, we use hand poses and 2D positions in the image space as surrogates for the supervised learning objectives. In the PoCo [83] dataset, we use 3D positions of the hand and use 6-Dof poses extracted by ICP as actions, and for EPIC-Kitchen [15], we use normalized 2D centers of the detection box as proprioceptions and the difference to the next frame as actions. We use in total of 2000 trajectories of video clips from EPIC kitchen with a maximum trajectory length of 500.

**Deployed Robot Dataset.** To further increase the heterogeneity in the pre-training dataset, we consider FrodoBot-2k [39] dataset that involves plays of driving robots in the wild for gaming. This dataset is composed of deployed mobile robots in the wild. We use the front camera for this dataset. The action space of this robot is parametrized by linear and angular velocity actions and the proprioception space includes measurements from the IMU. We use a total of 150 trajectories and each trajectory contains more than 500 steps.

See Figure 13 for visualization of some examples of heterogeneous dataset compositions. In practice when training on the mixture of these datasets, users can define customized sampling weights or apply stratified sampling methods. For generality, in this work, we use a balanced weight sampling method, commonly used in multitask learning. For these datasets, we process the visual features separately and save them on the disks.

## A.2 Network Details

**Stem.** For vision inputs, we resize each image to the standard square size (224x224) before feeding into a ResNet18 [26] into a 7x7 vision modality token. These tokens are specifically the features before the final global pooling. If multiple views are available, we create individual projector MLP for each image view and then concatenate the vision tokens. For the vision encoders. In Figure 18, We have experimented with multiple vision encoders such as MAE ViT base [24], Dino V2[59], and CLIP ViT Base [60]. We choose ResNet and the default image size for their simplicity and common

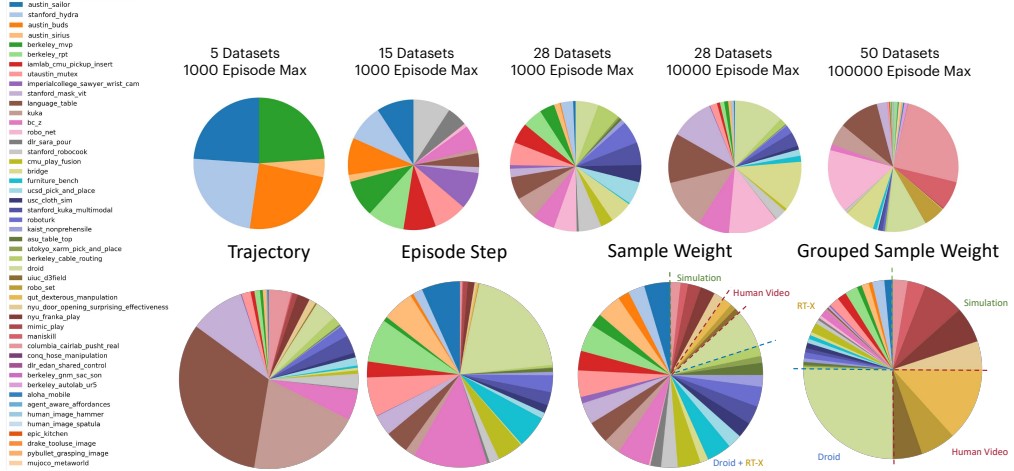

Figure 13: **Large-scale Dataset Heterogeneity in Robotics.** We show different dataset mixtures at increasing scales (top row) across trajectory counts, dataset sample counts, and sampling weights (bottom row). We also show illustrations of the different embodiments including real robots, simulations, and human videos. By default, during training, we use a uniform distribution to sample from each of the embodiment datasets.

usage in policy learning. The investigation of more complex fusion and processing for vision features is left to future works.

When language is used, we use T5 [64] to encode trajectory-level language modality tokens. Rather than using raw data and computing the tokens each time, these tokens are processed offline and saved as datasets before training.

For low-dimensional inputs such as proprioceptions and actions, we first flatten these vectors and then apply normalization to each dimension, as a single token. We have also experimented with increasing the dimensions of these modalities by adding sinusoidal position embeddings. For multiple steps in the observation horizon, we concatenate the sequence for each modality separately. At inference time, these tokens are forwarded once and cached to avoid multiple calculations.

We apply cross attention (with sinusoidal position encodings) and a shallow MLP projector [42, 46] within the stem to map different various sequences of tokens into a fixed number of tokens for that modality. The cross-attention layer has 8 heads and 64 hidden dimensions per head. We map the modality tokens into 16 tokens, 16 tokens, and 8 tokens respectively for image, proprioception, and language.

**Trunk.** The trunk is parametrized by a decoder-only transformer architecture with embedding dimension $h$ that we ablate from 64 to 2048, and with block numbers ranging from 16 to 80. Note that the number of parameters for the trunk scales quadratically with the dimension size and linearly with the number of layers. The trunk also supports loading from existing large language models. Refer to Table. 1 for more details on model sizes.

The code is modularized so that the trunk training and transfer are independent of the encoder architecture or pre-trained weights of the stem, which can be ImageNet pre-trained ResNet [26], R3M [54], Voltron [35], Dino v2 [59], etc, and can be fintuned during transfer learning. It is also independent of the head, which can be diffusion policies, MLP, or transformer.

**Head.** For the head architecture, we normalize the action spaces of each embodiment dataset to be between -1 and 1 element-wise, based on the dataset statistics, such that the output scale of the heads and the loss and gradients remain at similar scales. Since action trajectories and observation history can often help with the robotic problem, we pick a fixed action horizon (length 8) and observation horizon (length 4) for each embodiment. We then apply random masking along the time dimensions for each batch during training to be suitable for downstream tasks with different horizons.

| Dataset | Trajectory | Trajectory % | Sample | Sample % |
|---|---|---|---|---|
| Austin Sailor Dataset | 205 | 0.09% | 290167 | 1.85% |
| Stanford Hydra Dataset | 487 | 0.22% | 300217 | 1.92% |
| Austin Buds Dataset | 42 | 0.02% | 27894 | 0.18% |
| Austin Sirius Dataset | 478 | 0.21% | 235175 | 1.50% |
| Berkeley MVP | 410 | 0.18% | 34039 | 0.22% |
| Berkeley RPT | 776 | 0.35% | 326504 | 2.08% |
| IAMLAb CMU Pickup Insert | 539 | 0.24% | 118055 | 0.75% |
| UT Austin Mutex | 1283 | 0.57% | 295042 | 1.88% |
| Imperial College Sawyer Wrist Cam | 145 | 0.06% | 4519 | 0.03% |
| Stanford Mask VIT | 7788 | 3.49% | 155760 | 0.99% |
| Language Table | 29554 | 13.24% | 175855 | 1.12% |
| Kuka | 15903 | 7.13% | 73158 | 0.47% |
| BC-Z | 4365 | 1.96% | 559009 | 3.57% |
| Robo Net | 47127 | 21.12% | 895413 | 5.72% |
| DLR Sara Pour | 171 | 0.08% | 19462 | 0.12% |
| Stanford Robocook | 2103 | 0.94% | 73493 | 0.47% |
| CMU Play Fusion | 492 | 0.22% | 192988 | 1.23% |
| Bridge | 21768 | 9.75% | 500331 | 3.19% |
| Furniture Bench Dataset | 2763 | 1.24% | 2424703 | 15.48% |
| UCSD Pick And Place Dataset | 1158 | 0.52% | 45162 | 0.29% |
| USC Cloth Sim | 684 | 0.31% | 60876 | 0.39% |
| Stanford Kuka Multimodal Dataset | 2565 | 1.15% | 100021 | 0.64% |
| Roboturk | 1535 | 0.69% | 127523 | 0.81% |
| KAIST Nonprehensile | 171 | 0.08% | 25478 | 0.16% |
| ASU Table Top | 94 | 0.04% | 22029 | 0.14% |
| UTokyo Xarm Pick And Place | 78 | 0.03% | 4860 | 0.03% |
| Berkeley Cable Routing | 1266 | 0.57% | 19328 | 0.12% |
| Droid | 29437 | 13.19% | 2800000 | 17.88% |
| UIUC D3Field | 164 | 0.07% | 9803 | 0.06% |
| Robo Set | 15603 | 6.99% | 1042887 | 6.66% |
| QUT Dexterous Manipulation | 171 | 0.08% | 150698 | 0.96% |
| NYU Door Opening Surprising Effectiveness | 372 | 0.17% | 11418 | 0.07% |
| NYU Franka Play Dataset | 311 | 0.14% | 25536 | 0.16% |
| Mimic Play | 323 | 0.14% | 303738 | 1.94% |
| ManiSkill Dataset | 21346 | 9.57% | 2969893 | 18.96% |
| Columbia CairLab Pusht Real | 103 | 0.05% | 20375 | 0.13% |
| Conq Hose Manipulation | 96 | 0.04% | 4078 | 0.03% |
| DLR EDAN Shared Control | 88 | 0.04% | 6698 | 0.04% |
| Berkeley GNM SAC Son | 2526 | 1.13% | 183078 | 1.17% |
| Berkeley Autolab UR5 | 766 | 0.34% | 66425 | 0.42% |
| Aloha Mobile | 236 | 0.11% | 401754 | 2.57% |
| Agent Aware Affordances | 101 | 0.05% | 127403 | 0.81% |
| Epic Kitchen | 58 | 0.03% | 173012 | 1.10% |
| PoCo Hammer | 220 | 0.10% | 12517 | 0.08% |
| PoCo Spatula | 142 | 0.06% | 7517 | 0.05% |
| Drake Tooluse | 925 | 0.41% | 16650 | 0.11% |
| PyBullet Grasping Image | 1788 | 0.80% | 51852 | 0.33% |
| MuJoCo MetaWorld | 741 | 0.33% | 34725 | 0.22% |
| MuJoCo RoboMimic | 180 | 0.08% | 6735 | 0.04% |
| Isaac Arnold Image | 3214 | 1.44% | 16070 | 0.10% |
| PyBullet TriFinger | 147 | 0.07% | 89383 | 0.57% |
| MuJoCo Adroit | 90 | 0.04% | 8010 | 0.05% |
| FrodoBot | 60 | 0.03% | 12891 | 0.08% |

Table 4: **Detailed Dataset Mixture.** We include the detailed number of trajectories and the number of dataset samples in the training mixture. These include 41 dataset from Open-X [14], 7 datasets from simulation, 3 datasets from human video, and 1 from in-the-wild deployed dataset.

We support various types of policy heads in the network architectures such as standard MLP, transformer decoder, and diffusion policy. For the MLP head, we pooled the trunk feature (e.g. averaging) and then applied a 3-layer MLP. For the transformer decoder, we concatenate learnable tokens to the tokens before feeding into the trunk and use 1D convolution layer on these output tokens to regress actions. For diffusion policy in real world experiments, we use a diffusion head and train with DDPM [28]. Finally, the actions are unnormalized based on dataset statistics and Huber losses are applied for regression. The reason behind Huber loss is to balance between the "difficult frame" in robot trajectories and the easy but lengthy part of the trajectories.

As discussed, HPT is a meta-level architecture where the stem, trunk, and head code are modular and each supports various architectural changes with pre-trained weights. The inference time, which includes all the pre-processing and encoder time, on the local computer with an old NVIDIA RTX 3070 GPU is 39Hz for HPT-base and 33Hz for HPT-xlarge. A modern GPU such as A100 will likely improve the speed by 3-4 times.

### A.3 Pre-training Experiment Details.

We train HPT with AdamW [47] optimizer with a weight decay ratio 0.05, and a base learning rate of 0.0002 with a cosine learning rate schedule with warmups and dropouts. We apply proportional scaling of the base learning rate depending on the batch size. To support various horizons during

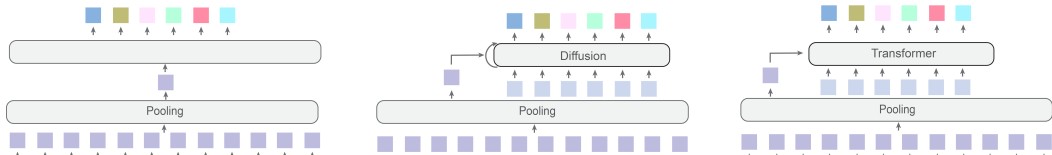

Figure 14: **Flexible Head Architectures in HPT.** We highlight that our HPT architecture is a meta-level architecture for policy learning, and it can work with various head architectures. The important takeaway is the scalable transformer architecture in the middle of the policy to absorb the diverse data and provide tokens for these policy heads to regress on the action outputs.

| Method | Dataset | Trajectories | Model Size | Heterogeneous Proprioception |
|---|---|---|---|---|
| RT-1 [6] | 1 | 0.1M | 16M | × |
| RT-2X [14] | 12 | - | 55B | × |
| Octo [56] | 25 | 0.8M | 93M | × |
| OpenVLA [36] | 25 | 1M | 7B | × |
| HPT | 52 | 0.2M | 1.1B | ✓ |

Table 5: **Experiment Statistics.** By leveraging heterogeneous datasets, the embodiment diversity in data and training scales reaches across several orders. Note that the training flops and the number of tokens are approximated from a single iteration and the model size only counts the trunk parameters (stem and head only have a small active parameter count. HPT is also provided with multiple open-source implementations and extensive simulation evaluation tasks across 6 different benchmarks.

transfer learning, we apply random masking along the time dimensions for each batch during training. Since action trajectories can have imbalanced losses along prediction horizons, we use Huber loss with $\delta = 0.1$ (empirically found). We found the pre-training stage to be stable across various hyperparameters in learning rate schedules and optimizers, and the choice of the validation dataset. The code is open-sourced and the pre-released model can be downloaded easily from Huggingface.

In practice, since training losses can vary across different datasets and our goal is to perform well on all embodiments and tasks, we apply a weighted sampling procedure for data loading. For every training iteration, we sample a dataset with inverse probabilities based on an exponential of its dataset size as a temperature. Specifically, we compute the squared root of each dataset size and sum these sizes to compute a normalization constant. For each batch item, we then sample from these dataset with the corresponding probability. This prevents large datasets from dominating a full training epoch, which is a common practice in multitask learning.

Note that the stem and head for each embodiment are updated in different frequencies than the trunk, similar to a mixture-of-expert [72] training procedure. Especially under distributed training settings, each stem and head is trained with data from a particular embodiment and tasks, and the trunk will accumulate gradients from all batches from training workers. The compute resources for these pre-training experiments range from 8 V-100s to 128 V-100s and the training time spans from half a day to 1 month. The total dataset disk size is around 10Tb and the RAM memory requirement is below 50Gb. See Table 5 for summary details of the experiment.

## A.4 Simulation Experiment Details

For simulation benchmarks, we use the released datasets as the expert demonstrations [90, 82, 13]. In summary, Metaworld [90] uses wrist camera view, and Robomimic [50] as well as Simpler [43] uses third-person view, with their own proprioception definitions by the dataset. Fleet-Tools [82] uses both views as inputs and uses the end effector poses as the proprioception inputs. We encode the image using pre-trained frozen ResNet features and normalize the proprioception inputs before passing them into the stem. We train single-task policies for all these simulation benchmarks except for Metaworld.

For the Simpler [43] benchmark, we focus on the `Close-Drawer`, `Move Near`, and `Pick Coke Can` task and Google EDR embodiment with a visual matching setting. We test 9 different initializations with a total of 218 episodes. Note that the simulation tasks have a focus on language conditioning and do not expose proprioception inputs, which is not the most suitable testbed for HPT. To address

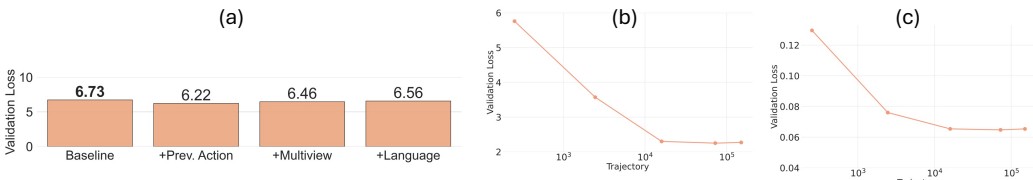

Figure 15: **Additional Architectural Ablation.** (a) We found that architecture changes on HPT-Base such as adding previous actions as inputs, multiview as inputs, and language input can help with HPT pre-training performance. (b,c) We ablate other policy head architectures such as discrete classification heads as well as action token heads by scaling along the number of trajectories. The experiment is conducted under the default setting with HPT-Base, fixed 27 datasets with 1000 max trajectories in each dataset.

these issues, we finetune HPT on the RTX supervised datasets with 79 trajectories as other simulation benchmarks. We use HPT-base as the backbone for this experiment.

By default, we train with 20000 iterations with batch size 512 and small learning rate $1e^{-5}$. The image and state stem are one-layer MLP with hidden dimension 128 and the head is two-layer MLP. We only use an observation window of length 1 and MLP as the policy head. Each training dataset uses from 10-100 trajectories per task and each test covers 50 episodes with different initial conditions. Each trajectory in the simulation has a slight difference in scene initialization. To reduce the variance, we conduct independent training runs 5 times, and the average for each baseline. In Figure 9, we show illustrations of some simulation tasks we evaluated.

### A.5    Real-World Experiment Details

**Task Definition.**    We experiment with robotic tool-use tasks `Sweep Leftover`, `Fill Water`, `Scoop Food` and `Switch Insertion` across two different robot setups. While for both setups, we use Franka Panda as the robot, we note that the sensor locations as well as the action spaces are drastically different. We collect approximately 100 demos for each task and evaluate each task for 15 trials to measure the average success rate.

During evaluation, a human supervises the robot at all times. An evaluation episode can be terminated due to safety concerns, robot faults, timeout, etc. An episode is considered successful if it accomplishes the task. In the `Fill Water` task, the success score of 1 means some water is poured into the bowl. In the `Sweep Leftover` tasks, a success score of 1 means all the piles are pushed into the plate, and a success score of 0.5 means some piles are pushed into the plate. In the `Scoop Food` task, a success score of 1 means some dog food is scooped and all is poured into the bowl and a score of 0.5 means some food is scooped. In the `Switch Insertion` task [91], a success score of 1 means the switch is precisely inserted into the three pins on the PCB board. The robot moves to a pre-defined pose before it attempts the insertion. We pick these challenging tasks as they require contact-rich interactions with tools and granular objects, and they require high-precision as well as dense contacts. We make sure the initial conditions of the robots are the same. Due to the complexity of these tasks and human errors, the initial conditions of the object setups are not exactly the same.

**Transfer Learning.**    For the policy head in the real-world experiments, we have experimented with both MLPs and diffusion policies [13]. Fine-tuning only has active parameters of no more than 3Mb, compared to much bigger models (e.g. 100M) often used for a single task in other works. We use an observation history window of 2 with a small learning rate $2e^{-5}$. We train with batch size 256 on a single NVIDIA RTX 2080Ti GPU for 20000 iterations (around 4 hours of wall time).

## B    Additional Experiments

In this section, we present some additional experiments and ablation studies.

### B.1    Additional Simulation Experiments

Based on the training curves of the four baselines in Figure 16 (a), we observe that leveraging a pre-trained HPT representation can often achieve a lower validation loss curve (lower) during fine-tuning.

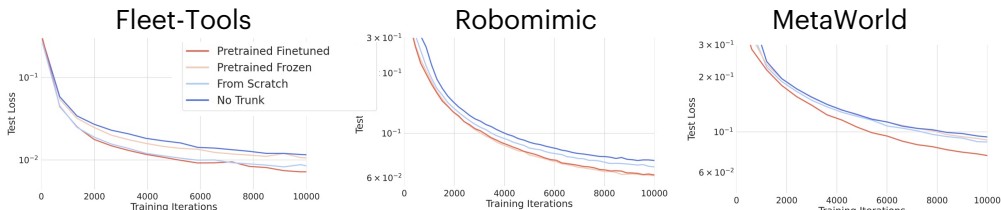

Figure 16: **Transfer Learning Objective.** We run transfer learning across several simulator benchmarks [82, 50, 90]. We compare the validation loss curves of several baselines with and without pre-trained HPT trunks. The pre-trained trunks are trained from the Default Settings.

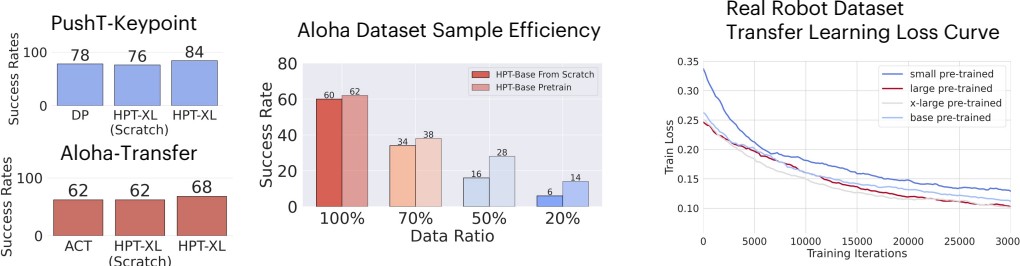

Figure 17: **Simulation Task Performance compared with Single-Task Policy in LeRobot Implementation.** We do evaluation in a different implementation in unseen simulation benchmarks. Left) we show that an improvement in performance can be achieved with pre-trained HPT trunks and outperforms single-task state-of-the-art architectures. We note that HPT trunks have not been pre-trained with diffusion heads and transformer decoder heads. Middle) we show the sample efficiency ablation study for HPT-Base. Right) We show model size ablation study in loss curves.

In Figure 10 (a), we run HPT on the Simpler [43] Benchmark, which allows for comparing with Octo [56], RT1-X, and RT2-X [14] on a high-fidelity simulation. We focus on three different tasks Close Drawer, Move Near, and Pick Coke Can in the Google EDR embodiment. For each task, we test several different initializations with a total of over 300 episodes for all tasks. Note that the pre-training corpus of HPT-S does not include [6], and simulation tasks have a focus on language conditioning and do not expose proprioception inputs, which is not suitable for HPT. To address these issues, we generate training data using the RT-1[6] as the expert, and finetune HPT on the RT-1X supervised datasets with around 50 trajectories and the simulation protocol. We use HPT-base with language tokenizers as the backbone for this experiment. We use the results from the paper [43].

Additionally, under the LeRobot [8] implementation, we compare HPT with state-of-the-art single-task policy architecture such as Diffusion Policy [13] and ACT [92]. In particular, we inherit a simple version of these complex policies as the head architectures (Figure 14). We run full training runs with 50 evaluation trials every 20000 training steps, and use the maximum task success rates during the 100000 training steps for comparisons. In Figure 17, we compare HPT with DP on the PushT task with keypoint representations using the diffusion head and achieve similar success rates at 78%. We also compare with ACT on the Aloha Transfer Box task with the image representations using the transformer decoder head and achieve similar success rates at 60%. This showcases the flexibility of HPT architecture to work with state-of-the-art policy architectures for high-frequency control in fine-grained tasks. Moreover, in the experiment with 50 episodes in total and HPT-base, we observe an improvement in sample efficiency with pre-trained models. We also see improvement in transfer learning loss with pre-trained models at increasing scales.

In Figure 10 (b), we ablate on the number of visual and proprioceptive tokens in the simulation transfer learning experiments. We observe that missing either information hurts the performance of the downstream policies. See Section A.4 for more implementation and experiment details.

### B.2 Ablation Study on the Stem

In this experiment, we fix the number of datasets (27) in Open-X datasets and use a maximum of 1000 trajectories for each dataset. We consider several ablations on the stem part of the HPTs.

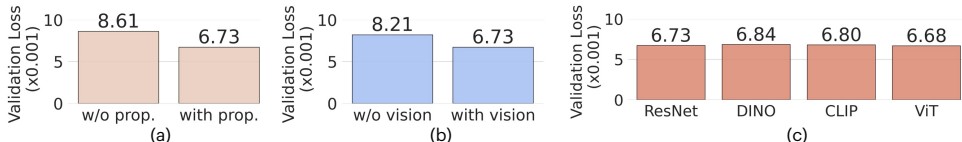

Figure 18: **Ablation Study on HPT Stem.** We ablate the pre-training performance for (a) proprioception, (b) vision stems, and (c) vision encoders. Setting: HPT-S, batch 256, iterations 80000, 27 datasets with a maximum of 1000 trajectories for each dataset.

(a) Test Initial Conditions          (b) Failure Cases

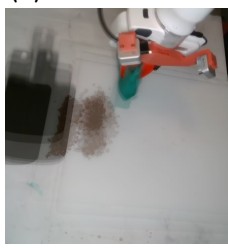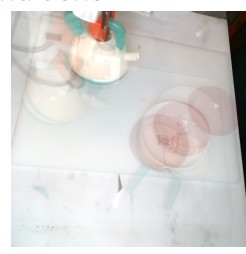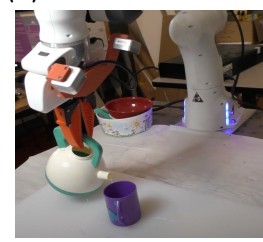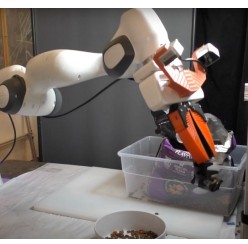

Figure 19: **(a) Initial Condition Overlay.** We visualize different rollout initial conditions during test times. **(b) Failure Cases of the Learned Policy in the Real World.** The robot sometimes has issues executing very precise manipulation.

Specifically, in Figure 18 (a, b), we observe an increase in validation losses when not using either the proprioception information or the vision. Intuitively, not using such proprioception or vision information would make learning action predictions from heterogeneous datasets very challenging. This also implies that both information are critical for policy pre-training at scale.

We also conduct an ablation experiment over vision backbones on a smaller subset of the training datasets among several popular vision encoders such as ViT-base [24] and DiNO [59] (Figure 18). Further ablating on input image resolution or joint finetuning of the vision backbone on the downstream task success rates will be interesting future work. Although the default implementation focuses on single-view visual information, the stem can naturally extend to multiple views and other modalities such as languages and action history.

### B.3 Pre-training Ablation Study

In Figure 15, we conduct several ablations to pre-train the HPTs, such as using multiple views to provide 3D information implicitly, using languages for task guidance, and using previous action trajectories as additional ablations. We found these ablations to improve over the default HPT setting with a single view and vision and proprioceptive inputs. These ablations are more pronounced in certain datasets such as multiple views for insertion [68], and language modality for Language Table [48]. Using previous action trajectories is helpful in providing additional context and embodiment information as well. We believe that integrating multiple modalities and investigating how to handle missing modalities is an exciting future direction. We leave the exploration of these ablations to future work.

From the architecture perspective, we also ablate over the token sizes and observation history and do not find a big effect on the averaged validation loss. We hypothesize there is a trade-off between the information given to the policy and the desired generalization. In Figure 15(b,c), we have also experimented with a discretized version [6] of the policy head and architecture that uses additionally learned position encodings as action tokens for transformer, with conv1D head for action regression. We opt for regression on continuous values for its generality. An initial investigation of the attention map of the transformer blocks shows that there is a dense attention weight attributed to the proprioception and vision tokens.

## C    Failure Cases

In Figure 19, we show some failure cases of the learned HPT policies in the real world. One of the common issues is overshooting or undershooting. For example, the policies tend to pour the water before it reaches the mug or pour the dog food a bit in front of the bowl. These issues could be due to the spatial precision of the policies and due to the data qualities failing to uncover the causal relationships. More targeted data recollection and better finetuning of the vision encoders might help address these issues.

## D    Discussion and Future Directions

We first discuss the metric used for measuring pre-training performance, or intrinsic evaluation. Validation (or training) loss is a common metric to evaluate the progress of large-scale pre-training [34]. It is based on the goal of training a generalist model, where even fitting all the data can be a challenge [58], as opposed to training a specialist model where overfitting is often appreciated. This is also considered a step towards more scientifically studying pre-training and scaling behavior in robotics [30]. Previously, people often rely on a single binary metric for evaluating task success rates in the real world, with only some amount of test trials.

Admittedly, there are several caveats to this metric. From the evaluation perspective, the averaged validation loss depends on the overall multitask losses of all datasets. For example, increasing the number of trajectories in one dataset might lower validation loss on the associated dataset, but may not lead to an overall loss decrease. In practice, the exact subset for evaluation and the number of training steps in each dataset also play a role in the averaged validation loss metric. For example, when evaluating whether additional datasets to the default setting contribute to the representation learning, selecting the default datasets allows us to compare on the same metric. But these datasets are inevitably trained less, if we fix the number of total training iterations. Moreover, the validation loss during pre-training and the downstream policy learning performance have an evaluation gap due to task differences, and downstream policy learning and task execution have another evaluation gap due to closed-loop execution.

Given the recent surge of scaled data, robot learning is still limited by its generality because of the heterogeneity, including different embodiments, tasks, and environments where the robots are operated. We propose HPT, a novel architecture and framework to embrace this heterogeneity through pre-training. We align proprioception and vision information of different embodiments through a modular stem, a shared scalable transformer trunk, and task-specific heads to actions. We explore and scale HPT with heterogeneous datasets to over 50 available datasets. The learned representation can be transferred and improved performance in both simulation and the real world. We hope this perspective will inspire future work in handling the *heterogeneous nature* of robotic data.

Since fine-tuning is still required for robotics generalist models, future research directions include exploring different algorithms including network architectures that incorporate embodiment-specific structures, such as URDF, into network architecture as well as tokenization. One can also explore training objectives beyond supervised learning. As we move towards scaling robot pre-training, more high-quality diverse datasets with clear annotations would be crucial, including teleoperation data, simulation data, human videos, and deployed robot data. Understanding what kinds of mixture will contribute to better representations is interesting future work.

Due to the complexity of real-world evaluation, large-scale unified simulation benchmarks with varying dexterity and generalization challenges would be very crucial to consistently compare among different models. For real-world policy performance, we believe that extending to longer-horizon fine manipulations with a bimanual or mobile setup would be interesting future works.

Future works can also study and extend scaling laws in policy learning, and explore representations from heterogeneous data in other domains beyond robotic policy learning. Finally, leveraging more modalities and domains in robotics such as 3D point clouds, tactile data, simulation domains, and human data, etc, is worth investigating.

