# OpenReview forum: "Scaling Proprioceptive-Visual Learning with Heterogeneous Pre-trained Transformers"
_NeurIPS.cc/2024/Conference — NeurIPS 2024 spotlight_

### Official Review · Reviewer_KFGf · 2024-07-09

**Soundness:** 2
**Presentation:** 3
**Contribution:** 3
**Rating:** 6
**Confidence:** 4

**Summary:**

The paper introduces HPT a large-scale transformer model pretrained on multi embodiment robotic data. The main idea is to split the policy into three parts: a dataset specific token encoder for images and proprioceptive information called Stem. A shared Trunk, that processes all latent tokens and a dataset specific action head. By splitting the architecture into dataset specific encoders and dataset agnostic trunk the model is able to better process data from different robot embodiments and scale well using diverse data from Open-X-Embodiment Dataset (OXE). The proposed architecture is tested on pretraining on OXE and adapted for sim and real robot experiments.

**Strengths:**

- Simple but yet effective method that shows good scaling properties for a general multi-embodiment robot transformer.
- Large scale pretraining study covering 50 datasets and many robot embodiments. Most diverse pretraining in the context of robotics that I am aware of. The reported results shows good potential for further exploring heterogeneous robot data sources
- Insightful scaling analysis, which do not exist in the context of robotics yet.
- Several additional ablation studies to further study the impact of future analysis with discrete action spaces more input data and first results for fleet learning experiments underline the potential of the architecture.
- The paper is well structured and easy to follow.

**Weaknesses:**

- No adequate baselines in the real world: A comparison of a model pretrained on full robot trajectories with action prediction against a few pretrained vision encoders that are mostly trained on robotic-free first person human videos is not a fair or useful comparison and does not provide relevant insights for the readers. At least, I expect a comparison against Octo [1] as another generalist policy pretrained on OXE. Also the experiment is missing baselines trained from scratch such as Diffusion Policy or ACT. Further the experiments in the real world only cover a single robot embodiment.
- For a method that is motivated by better learning from heterogeneous robot data, some additional results with other robots having different action spaces would be required to showcase the advantages of the proposed method
- No adequate baselines in simulation: The experiments on robotic datasets such as Robomimic and the real world again do not contain adequate comparisons. Specifically in the Robomimic experiment there is no comparison against state-of-the-art in-domain Diffusion Policy or ACT trained from scratch. The are no insights if the proposed model pretrained on diverse data improves upon these models trained from scratch. While comparison against the same architecture with different ablations exist, comparisons against other architecture are missing. Experiments on the recently proposed SimlerEnv [2] could also be interesting to compare the performance of the method against other large-scale pretrained methods such as Octo and RT-2X.
- Inference time: While the performance of larger variants of HPT is promising, I am not convinced by the idea that an $80$ layer transformer model is a suitable backbone for a policy used in a real robot context.
- Several points about the image-encoder are not well described and miss relevant details.

[1]: Team, Octo Model, et al. "Octo: An open-source generalist robot policy." _arXiv preprint arXiv:2405.12213_ (2024).

[2]: Li, Xuanlin, et al. "Evaluating Real-World Robot Manipulation Policies in Simulation." _arXiv preprint arXiv:2405.05941_ (2024).

**Questions:**

- Can you add adequate baselines such as Octo for the related experiments in sim and real to better justify the performance of HPT as mentioned in the weakness section.
- I found the part about the image encoder a bit short of details. Is the ResNet pretrained on Image-Net in the final model or which pretrained image encoder are you using? Have you experimented with also fine-tunining it? Since ViT has a lower validation loss why did you decide to use ResNet as the default? Which pre-trained Vit are you using for the ablation? More details on this part would be helpful for the reader.
- How fast is the inference time for the different models?

**Limitations:**

Yes

---

> ### Author Rebuttal · Authors · 2024-08-04
>
> Thank you so much for the extensive feedback and suggestions. We have revised the manuscript and **conducted new experiments in the attached PDF**  and will address your questions below.
>
> > No adequate baselines in the real world .... I expect a comparison against Octo [1] as another generalist policy pretrained on OXE. Also the experiment is missing baselines trained from scratch such as Diffusion Policy or ACT. Further the experiments in the real world only cover a single robot embodiment.
>
> We have added additional experiments to compare against several generalist policies in the recently released Simpler Benchmark. See Figure 13(a) for the result.  Specifically, we compare the models across 3 different tasks in the Google GDR embodiment with the visual matching setup. We have found HPT to perform quite well on these benchmarks compared to Octo for instance. For the real-world experiments, unfortunately, We have not been able to make Octo finetuning work well in our real-world setup despite weeks of effort, so its current finetuning performance does not match really well. It can be trained on the local machine but the trained policies are not doing anything meaningful. We hypothesize that there might be issues in the finetuning process.
>
>
> The Fill-Water task in the real-world experiment in **Figure 14,15** is actually conducted with the **diffusion policy** architecture. Therefore the from-scratch baseline represents state-of-the-art single-task diffusion policies. We use the relative policy action output for the diffusion policies. We observe an improvement by using the pre-trained trunk compared to the train-from-scratch baselines. We highlight again that the ACT or diffusion policies are only considered heads of specialist policy in this framework, so it’s orthogonal to our contribution and is only used to showcase the flexibility of the pre-trained representations to improve model performance. In the Appendix, we also investigate the difference between using different heads in pre-training. We will clarify these in future versions of the writing.
>
> > For a method that is motivated by better learning from heterogeneous robot data, some additional results with other robots having different action spaces would be required to showcase the advantages of the proposed method.
>
> Thanks for bringing this up. We have added additional experiments on a different embodiment conducting real-world insertion tasks. The action space of this robotic setting is absolute pose, rather than the relative pose used in the pet care setting. Please **Figure 14 and Figure 15 left** for more details and visualizations. We hope this answers your question.
>
> > Inference time: While the performance of larger variants of HPT is promising, I am not convinced by the idea that an 80 layer transformer model is a suitable backbone for a policy used in a real robot context. How fast is the inference time for the different models?
>
> The inference time, which includes all the pre-processing and encoder time, on the local computer with an **ancient** NVIDIA RTX 3070 GPU is 39Hz for HPT-base and 33Hz for HPT-xlarge. It is expected to be 3-4 times faster on a more modern A100 used in OpenVLA. Indeed, the largest model can sometimes be overkill for local deployment of robot policies. But the HPT-XL model can easily run in real-time with our limited hardware in our real-world experiments.
>
> > Can you add adequate baselines such as Octo for the related experiments in sim and real to better justify the performance of HPT as mentioned in the weakness section.
>
> We have added additional experiments to compare against several generalist policies in the recently released Simpler Benchmark. See Figure 13(a) for the result. Specifically, we compare the models across 3 different tasks in the Google GDR embodiment with the visual matching setup. We have found HPT to perform quite well on these benchmarks compared to Octo for instance. For the real-world experiments, unfortunately we have not been able to make Octo finetuning work well in our real-world setup despite weeks of effort, so its current finetuning performance does not match really well. It can be trained on the local machine but the trained policies are not doing anything meaningful. We hypothesize that there might be issues in the finetuning process.
>
> > I found the part about the image encoder a bit short of details ..... More details on this part would be helpful for the reader. Several points about the image-encoder are not well described and miss relevant details.
>
> Thanks for bringing this up. We have added additional details to the network section in the appendix. We have copied it here for completeness. For vision inputs, we resize each image to the standard square size (224x224) before feeding into an Imagenet pre-trained ResNet18 into a 7x7 vision modality token, which are the features before the final pooling layer. If multiple views are available, we create individual projector MLP for each image view and then concatenate the vision tokens. We use ResNet mostly for its simplicity and legacy reasons (as the majority of the dataset is only processed once at the beginning of the project).
>
>
> Due to the computation resource limitation and other practical reasons, we did not finetune it as in other VLM works by default. We were using the ViT-base from the masked autoencoder and CLIP for vision encoder ablation. We hypothesize that the vision features for robotic control do not differ by that much across these features learned from the standard images found in vision benchmarks or the internet. We think exploring this further can be interesting future work.
>
> Thank you very much again for your constructive feedback. Please do not hesitate to let us know if you have any other questions and/or comments.

---

> > ### Comment · Reviewer_KFGf · 2024-08-08
> > **Thank you for the detailed answers - some followup questions**
> >
> > I would like to thank the authors for their thorough and thoughtful responses to my concerns. The new experiments and details will significantly strengthen the paper.
> >
> > However, I have a few remaining questions:
> >
> > - Is there a reason for not testing on other Simpler Tasks? As far as I know what HPT trained on both datasets, so it could be interesting to include the full results on the bridge tasks too for the final version of the paper. This comprehensive reporting could serve as a valuable baseline for future research in the field of foundation models.
> >
> > - Training Code: While it's good that the pretrained models will be open-sourced, I'm curious about the plans for releasing the training pipeline. Making this available could greatly benefit the robotics community.
> >
> > - Proprioceptive Input: Could you elaborate on why this approach was effective in your model, particularly in light of potential overfitting or causal confusion issues that have been observed in other works like Octo and also a lot in my personal experiences?

---

> > > ### Author Response · Authors · 2024-08-08
> > > **Response to Follow-Up Questions**
> > >
> > > > Is there a reason for not testing on other Simpler Tasks? As far as I know what HPT trained on both datasets, so it could be interesting to include the full results on the bridge tasks too for the final version of the paper. This comprehensive reporting could serve as a valuable baseline for future research in the field of foundation models.
> > >
> > >
> > > Thanks the reviewer for bringing this up. We were running out of time when preparing this rebuttal. We will try to add the bridge tasks for the final version of the paper and to provide a baseline for future research.
> > >
> > >
> > > > Training Code: While it's good that the pre-trained models will be open-sourced, I'm curious about the plans for releasing the training pipeline. Making this available could greatly benefit the robotics community.
> > >
> > >
> > > Thanks the reviewer for bringing this up. Yes, following the great practice of Octo and OpenVLA, the entire codebase for the training pipeline and pre-trained models will all be open-source for the robotics community upon acceptance. We would love to see how the community will use our models as well as any feedback and potential improvements.
> > >
> > >
> > > > Proprioceptive Input: Could you elaborate on why this approach was effective in your model, particularly in light of potential overfitting or causal confusion issues that have been observed in other works like Octo and also a lot in my personal experiences?
> > >
> > > We first note that proprioception provides critical information for achieving human-like dexterous motions, especially in high-dimensional control such as dexterous hands and legged locomotion. Without appropriate architecture, extensive training, and appropriate data filtering, indeed there can be overfitting or causal confusion between the actions and proprioception. However, in this work, the proprioception is early-fusion with the vision information, both of which take equal amounts of tokens (16), which makes proprioception information usage different from the traditional usage: feature concatenation with the low-dimension vector after feature processing for the vision. We hypothesize that this design with large scale heterogeneous training might be the reason why proprioception contributes to our findings, as observed in Figure 5 and Figure 13.
> > >
> > >
> > > Thank you very much again for your constructive feedback. Please do not hesitate to let us know if you have any other questions and/or comments.

---

> > > > ### Comment · Reviewer_KFGf · 2024-08-09
> > > > **Thanks for the answer. More follow-up questions.**
> > > >
> > > > Thanks for the answers! I increased my score to 6.
> > > >
> > > > I have some further questions:
> > > >
> > > > - Could you elaborate on the action spaces of the real world experiments you added for the rebuttal?
> > > >
> > > > - Do you have any insights on fine-tuning vs Co-Training on your model that you could share? Which variant is better for your model? Given the modularity of your approach I am curious is Co-Training is the upper bound for your model too.

---

> > > > > ### Author Response · Authors · 2024-08-09
> > > > > **Response to Follow-Up Questions (Continued)**
> > > > >
> > > > > > Could you elaborate on the action spaces of the real world experiments you added for the rebuttal?
> > > > >
> > > > > The action space for the switch insertion task is position control, as opposed to the velocity control in the pet-care tasks. It works as follows: The robot takes in the current end effector translation and observed image and input, and output the goal end effector translation that the robot needs to reach. The robot controller is an impedance controller that is tuned to executed these fine-grained insertion motions with compliance.
> > > > >
> > > > >
> > > > > > Do you have any insights on fine-tuning vs Co-Training on your model that you could share? Which variant is better for your model? Given the modularity of your approach I am curious is Co-Training is the upper bound for your model too.
> > > > >
> > > > > This is an excellent question. We believe that in the ideal case with enough engineering, co-training will work better than finetuning, as it circumvents the catastrophic forgetting issues in finetuning and allows for the full network to update. Indeed, as you pointed out, the modularity in our approach will create a new stem/head for a new domain, independent of finetuning or co-training. Therefore, the main difference lies in whether the trunk gets to share the training with other datasets and gets to update during adaptations. In our experiments, the joint training regime will work better, but a more focused sampling for the target dataset is necessary, because a balanced sampling regime might lead to under-training for the adapting domain.
> > > > >
> > > > >
> > > > > Thank you very much again for your constructive feedback and the score raise. Please do not hesitate to let us know if you have any other questions and/or comments.

---

### Official Review · Reviewer_wUp5 · 2024-07-12

**Soundness:** 3
**Presentation:** 3
**Contribution:** 2
**Rating:** 3
**Confidence:** 4

**Summary:**

This paper presents Heterogeneous Pre-trained Transformers (HPT), a method for training robotic models that addresses the challenge of heterogeneity across different robot embodiments and tasks. HPT pre-trains a shared neural network trunk to create a universal representation, which is then fine-tuned for specific tasks, showing improved performance and scalability with diverse datasets as well as demonstrating effective generalization to new tasks in both simulation and real-world settings.

**Strengths:**

This paper shows that pre-training policies across heterogeneity can exhibit scaling behaviors of validation losses.

This paper is well-written and easy to understand.

**Weaknesses:**

Despite the difficulty of large-scale evaluation, there is no inevitable relationship between the loss values and the task success rates. For instance, the main challenge in some tasks lies in certain key frames, rather than the average error in lengthy trajectories.

In a sense, verifying the scaling behavior at the level of loss values is quite evident.

In terms of architectural design, this paper (stem, trunk and head) does not offer significant technical contributions.

Lacking comparisons with baselines (only some vision encoders on Sweep Leftover Task), such as RT-X and Octo.

In Figure 5, why is the loss lower without vision than without proprioceptive information? I am puzzled. Does "no vision" here mean that the policy has no visual information as input? Why can a reasonable loss value still be obtained without vision?

**Questions:**

N/A

---

> ### Author Rebuttal · Authors · 2024-08-04
>
> Thank you so much for the extensive feedback and suggestions. We have revised the manuscript and **conducted new experiments in the attached PDF** and will address your questions below.
>
> > Despite the difficulty of large-scale evaluation, there is no inevitable relationship between the loss values and the task success rates. For instance, the main challenge in some tasks lies in certain key frames, rather than the average error in lengthy trajectories.
>
> Thank you for bringing this up. Indeed, we use the smooth L1 loss to balance between the “difficult part” and the easy but lengthy part of the trajectories. I agree that it’s not optimal to use the validation loss and therefore we conduct many transfer learning experiments and only use the validation loss to evaluate the pre-training performance. Please see the paper for more discussions on using this metric. Since the submission, we have also conducted new experiments on a new real-world embodiment and did additional experiments in simulation benchmarks.
>
> **We also copy the general response here for completeness.**
>
> We highlight that the experiments in this work are three-fold: pre-training, evaluation in the simulation benchmark (**Figure 16**), and evaluation in the real world (**Figure 14, 15**). Only in the pre-training part, where we need to extensively measure **the scaling behaviors of the learned representations**, do we use the validation losses. It’s not an ideal choice, but it’s a common choice adopted by foundation model training in both LLM and multimodal foundation models. We note that the standard practice of robot learning (specialist model) is to overfit on ~100 trajectories, in which cases validation loss might not be that indicative, and evaluation is not that difficult. But in our case, we are training representations on massive amounts of robot embodiments and trajectories (generalist model), in which faithful evaluation of such representation without expensive rollouts is still an open question. Compared to earlier works that ignore this metric, we believe validation loss is one step towards that goal and should be representative of the learning progress in this process than single-task training. We also investigate their correlations with transfer learning performance in later experiments, where we show that bigger and more well trained models indeed perform better.
>
> Additionally, we have proposed a simple fix (using smooth L1 loss) to make it more balanced between the difficult and easy parts of the trajectories. This does not include the high costs of evaluating representations in robotics. Previously, validation losses were simply ignored by practitioners, and our work is one step towards that direction. How to evaluate robot representations without running a large amount of rollouts is still an open research question. We believe that validation loss can be a more continuous reflection of representation at a large scale, and we will further revisit and investigate the correlations at a large scale in future work.
>
> Last but not least, we did many experiments in the two sections on transfer learning in simulation and the real world. The simulation experiment is more extensive than previous works with thousands of runs. These preliminary experiments have also showcased the correlation of validation loss and transfer learning performance (**Table 4 and Figure 16b**). We want to highlight that previous works pre-trained on Open-X have evaluated mostly using the training embodiments, which we do not have access to. Our real-world evaluation tasks are also more challenging than previous works.
>
> > In a sense, verifying the scaling behavior at the level of loss values is quite evident.
>
> Thank you for bringing this up. This scaling behavior analysis has become a norm in the LLM domain, but in robotics, there has not been any work investigating similar problems to the best of our knowledge. Under such heterogeneous training across diverse embodiments and tasks, it is unclear to us beforehand if the training will even converge or proceed without instability issues. We will highlight the part on model transfer learning in the revised version.
>
> > In terms of architectural design, this paper (stem, trunk, and head) does not offer significant technical contributions.
>
> Thank you for bringing this up. Indeed, the architecture was designed from a simple and general mindset to tackle the problem of heterogeneous learning. It is different from a monolithic model such as Octo and OpenVLA, in order to handle the heterogeneity in embodiments, and is also aligned with the multimodal learning literature. Besides the heterogeneous pre-training and architecture contributions, our work also provides insights into the scaling behavior investigations and conduct evaluation in simulations. (continued)

---

> ### Author Response · Authors · 2024-08-04
> **Response to Review of Submission7169 by Reviewer wUp5 (Continued)**
>
> We also copy the general response here for completeness.
>
> - We highlight that the main motivation to tokenize embodiments and align them in the same latent space, is due to the practical constraints of heterogeneity in **proprioception** and action spaces. These input and output tensors with different dimensions cannot be handled with a **single network** technically (think about two weight matrices with different dimensions), and are **not considered** in previous works before. To handle these, we use a lightweight encoder-decoder approach in this project. The idea is partly motivated by the “experts” in the recent mixture of expert literature as well as the “projector” or “connector” in the multimodal literature. For the vision modalities, we actually **share** the vision encoder and only use a small adaptor to project the frozen vision encoders.
>
> - Therefore, this is also a meta-level choice that is brought by heterogeneous pre-training, and is independent of specific architecture choices (shared head or separate heads). While this HPT architecture is a specific architectural choice that we made, the idea is motivated to handle **more heterogeneity across domains and sensor modalities**. This is a more **significant point and is different** than previous works. There are more complex methods such as using GNN or a unified tokenizer across these embodiments to handle this issue, but we leave those for future work.
>
> -  Moreover, this simple and general architecture choice is **not** conflicted with sharing information across embodiments because the encoders (stem) and decoders (head) do not contain many parameters, and are merely applied to match the dimensions and project tensors in different spaces to the shared latent space, similar to how multimodal foundation models are trained. Moreover, in all robotic generalist works including Octo and OpenVLA, finetuning is still necessary,  which means some information in the new embodiment is not learned during pre-training.
>
> > Lacking comparisons with baselines (only some vision encoders on Sweep Leftover Task), such as RT-X and Octo.
>
> We have added additional experiments to compare against several generalist policies in the recently released Simpler Benchmark. See Figure 13(a) for the result.  Specifically, we compare the models across 3 different tasks in the Google GDR embodiment with the visual matching setup. We have found HPT to perform quite well on these benchmarks compared to Octo for instance. For the real-world experiments, unfortunately we have not been able to make Octo finetuning work well in our real-world setup despite weeks of effort, so its current finetuning performance does not match really well. It can be trained on the local machine but the trained policies are not doing anything meaningful. We hypothesize that there might be issues in the finetuning process.
>
>
>
> > In Figure 5, why is the loss lower without vision than without proprioceptive information? I am puzzled. Does "no vision" here mean that the policy has no visual information as input? Why can a reasonable loss value still be obtained without vision?
>
> Depending on the tasks, there could be coupling information from the proprioceptive input and the action output. Our hypothesis is that in some simpler tasks, merely memorizing proprioceptive history and doing extrapolation leads to better performance compared to vision-only systems. Therefore, even if no vision information is used, there can still be a loss value. We have added an additional experiment in Figure 13(b) to show in transfer learning experiments in simulation, vision tokens are still necessary to achieve good performance measured by task success rates.
>
> Thank you very much again for your constructive feedback. Please do not hesitate to let us know if you have any other questions and/or comments.

---

> > ### Author Response · Authors · 2024-08-09
> > **Response to Review of Submission7169 by Reviewer wUp5 (Continued)**
> >
> > Dear Reviewer, we would like to thank you again for your efforts and time in providing thoughtful feedback and comments. We’ve conducted additional experiments according to your suggestions and replied to all the questions and concerns. Since the discussion period is ending soon, we would greatly appreciate it if you could let us know whether you have any additional comments. Thanks a lot!

---

> > ### Comment · Reviewer_wUp5 · 2024-08-12
> >
> > I have read the authors' responses and appreciate their efforts. However, I still have some concerns:
> >
> > 1. In transformer-based robot models, I believe that what is more challenging to scale up than training data and model capacity is the scale of evaluation. Before RT-X and Octo, there have already been lots of works on verifying the scaling laws of robotic learning. Compared with these works, I hope to see more in-depth experiment findings, such as which data are more important, rather than just finding that the model can scale up.
> >
> > [1] A Generalist Agent. TMLR 2022
> >
> > [2] Vima: General robot manipulation with multimodal prompts. ICML 2023
> >
> > [3] SMART: SELF-SUPERVISED MULTI-TASK PRETRAINING WITH CONTROL TRANSFORMERS. ICLR 2023
> >
> > [4] RT-1: ROBOTICS TRANSFORMER FOR REAL-WORLD CONTROL AT SCALE. RSS 2023
> >
> > [5] Robocat: A self-improving foundation agent for robotic manipulation. 2023
> >
> > 2. In addition, I agree with review Vi4L's point that I believe it is inappropriate to use the validation set loss for a large number of experiments as a substitute, because the policy is multimodal, or the authors can ensure that the data in the validation set can well cover various situations. Changing the loss function will not solve this problem.
> >
> > 3. As the authors mentioned, the architecture design of this paper is merely aimed at addressing the dimension alignment problems among different embodiments. However, some works such as SMART [3] also deal with different embodiments, using shared prediciton head and tokenizer during pre-training. I would like to know what findings and conclusions the authors have made in terms of model design?
> >
> > 4. I still cannot understand why the model can achieve a 34% success rate without visual information input. Could the authors elaborate on the details of the fine-tuning and evaluation setup? For example, what are the differences between fine-tuning and testing.
> >
> > 5. As for different datasets, what modalities are included in HPT's input? What types of task instructions are there?

---

> ### Author Response · Authors · 2024-08-12
> **Response for Reviewer wUp5**
>
> Thank you so much for the new feedback and suggestions. We will address your questions below.
>
> > 1. In transformer-based robot models, I believe that what is more challenging to scale up than training data and model capacity is the scale of evaluation. Before RT-X and Octo, there have already been lots of works on verifying the scaling laws of robotic learning. Compared with these works, I hope to see more in-depth experiment findings, such as which data are more important, rather than just finding that the model can scale up.
>
> Thanks reviewers to bring these great works up and we will add these works to the reference. Indeed, the study on using transformer for pre-training has a line of research. In addition to real world and simulation transfer learning, we propose to use the large-scale validation losses to measure pre-training and scaling law similar to fields such as LLM, precisely because of the "challenges in the scale of evaluation" as you mentioned. What’s new after RT-X is the open-source large-scale heterogeneous data in the real world. Therefore, the main focus of this work is to leverage such real world expensive data, and intuitively we believe such data has less domain gaps to real world transfer learning and are considered more precious. At the same time, such data also presents huge heterogeneity in terms of the tasks and the embodiments. This is also one of the main difference from earlier work such as Gato and VIMA. For example, the real world training data can include mobile manipulation for dexterous tasks and large-scale language instructions for table-top settings.
>
> In terms of data experiments in the paper, we have conducted experiments in scaling dataset number, data trajectories, and different domains of data including simulation and human videos. We have some findings such as increasing the dataset number can improve the trunk representations, despite its large distribution shifts. We discover that adding more trajectories (often >1000) for one specific embodiment such as in the Google Kuka dataset can often lead to marginal improvements, while the diversity of the pre-training datasets is still limited.
>
> The joint training performance improvement also applies to simulation and human videos, which are under-explored in the community. In the ablation, we have also investigated and presented the training performance of each specific dataset in the mixtures. We discover that there are several modes of learning, which might be due to the difficulties of that embodiment and task:  For certain tasks, the loss can also improve suddenly during training (grokking) and for some datasets, it can steadily improve or plateau. We will add more discussions on the data findings to the manuscript.
>
>
>
>
> > 2. In addition, I agree with review Vi4L's point that I believe it is inappropriate to use the validation set loss for a large number of experiments as a substitute, because the policy is multimodal, or the authors can ensure that the data in the validation set can well cover various situations. Changing the loss function will not solve this problem.
>
> We have discussed extensively the rationale and motivations for using validation losses for **evaluating pre-training**. Please see our common answer and manuscript for details. Each validation dataset contains up to 100 trajectories which covers a wide range of test cases. As mentioned earlier, the scale of evaluation is rather challenging, so we are also open to other suggested metrics for evaluating pre-training performance. We have conducted additional transfer learning experiments to verify the correlations between representation and its downstream task performance.
>
> Thanks again for bringing this up. In our newly revised manuscript, we reorganize the content such that the paper focuses more on transfer learning (Section 5).
>
> (to continue)

---

> > ### Author Response · Authors · 2024-08-12
> > **Response for Reviewer wUp5 (continued)**
> >
> > > 3. As the authors mentioned, the architecture design of this paper is merely aimed at addressing the dimension alignment problems among different embodiments. However, some works such as SMART [3] also deal with different embodiments, using shared prediciton head and tokenizer during pre-training. I would like to know what findings and conclusions the authors have made in terms of model design?
> >
> > Indeed SMART is a seminal and relevant work that also studies generalist policy, and we will add it to the reference. However, they have a slightly more targeted focus on RL with pure simulation experiments on deepmind control (DMC), originated from decision transformer. The main difference from our work is that we focus on heterogeneous pre-training with more challenging real world tasks, which includes 52 different datasets real world, simulation, and human videos. We also validate our findings in the real world transfer learning experiments. We also did not apply causality, masking, or dynamics learning in the pre-training process. For example, real world experiments demand much higher generality across RGB images by leveraging vision, as compared to simulation experiments in locomotion.
> >
> > In terms of the model design, we highlight it’s **”not merely aimed to addressing the dimension alignment problems”**. We propose stem network that encodes various modalities into a shared latent space throught **cross-attention**. Because of the drastically different embodiments in the complex real world scenarios, we use **different stems and heads** for different embodiments, as opposed to “shared prediciton head and tokenizer”. In terms of findings, we ablate over different kinds of input modalities in this network architecture. For example, we find that adding language instructions and multiple views can sometimes help with the performance. We also have run extensive experiments in scaling data, training schedule, and model sizes, which are useful insights for future works.
> >
> >
> > > 4. I still cannot understand why the model can achieve a 34% success rate without visual information input. Could the authors elaborate on the details of the fine-tuning and evaluation setup? For example, what are the differences between fine-tuning and testing.
> >
> > For that particular experiment using Fleet-tools on Figure 13, we adopt the HPT finetuning and evaluation procedure to reinitialize the head and stem, and train on 200 demonstrations generated by a motion planner, and we then evaluate on the unseen scenes. The robot starts at a fixed initialization and there are object initialization randomness between fine-tuning and testing. But the variation is not very significant, which is why there are non-zero success rates even without visual inputs.
> >
> > > 5. As for different datasets, what modalities are included in HPT's input? What types of task instructions are there?
> >
> > We mainly use vision and proprioception as the input modalities for HPT. We can optionally use language instructions as the inputs. In the open-x embodiment datasets, there are various sorts of task instructions from high-level instruction for the entire task such as “route cable” in berkeley_route_cable dataset to low-level motion instructions such as “grasp the yellow block and turn it left” in taco_play dataset.
> >
> >
> >
> > Thank you very much again for your constructive feedback. Please do not hesitate to let us know if you have any other questions and/or comments.

---

> > > ### Comment · Reviewer_wUp5 · 2024-08-13
> > >
> > > Does this mean that HPT has no instruction/condition at all? I believe a multi-task policy should at least need a goal image as a condition. I'm sorry that I did not find these details in the paper.

---

> ### Author Response · Authors · 2024-08-13
> **Response to Reviewer wUp5**
>
> As previously discussed, HPT pre-training is capable of handling multiple modalities, including images, proprioception, and language. For tasks that require specific conditions, such as those in the Simpler Benchmark, we include language instructions. While we did not explicitly explore using goal images as conditions and we do not think this is necessary for all robotic tasks such as single-object grasping, this could be another effective way to convey task information, potentially offering more detailed guidance than language instructions. Other than pre-training time, new modalities can also be added and feature backbones can be finetuned during transfer time thanks to the global pooling of the trunk.
>
> Please do not hesitate to let us know if you have any other questions and/or comments.

---

> > ### Author Response · Authors · 2024-08-13
> > **Response to Reviewer wUp5 (Continued)**
> >
> > Hi Reviewer wUp5,
> >
> > We hope that we have addressed most of your concerns and questions. Since this is the last day of the rebuttal, please don't hesitate to let us know if you have any other final questions and/or comments. Thanks!

---

### Official Review · Reviewer_AQbN · 2024-07-12

**Soundness:** 3
**Presentation:** 3
**Contribution:** 3
**Rating:** 7
**Confidence:** 4

**Summary:**

This paper aims to pretrain a policy representation across a variety of simulated and real robotics datasets such that it can be quickly adapted to new downstream problem instances (embodiment, environment, task). The key technical contribution is the design of a Transformer architecture that can consume a variety of robot data (i.e., observations and actions unique to each robot dataset) via a large sharable trunk + a number of small networks that translate dataset-specific information to this shared representation space. Experimental results indicate that the proposed architecture scales both in terms of dataset size / diversity and model capacity, and that it can be transferred to new real-world problems by finetuning on relatively few demonstrations (<=100).

**Post-rebuttal update:** I believe that my concerns have been addressed by the author rebuttal. I was already leaning towards acceptance and have now updated my score from 6 -> 7.

**Strengths:**

- Problem is interesting, timely, and likely to be of relevance to the NeurIPS community.
- Paper is well written and easy to follow. I believe that there is sufficient discussion of related work for unfamiliar readers to appreciate the technical contributions.
- The proposed architecture is fairly simple and intuitive. I consider this a strength. A series of ablation experiments validate the approach.
- Limitations of the evaluation metric (validation loss) is clearly stated. I don't think anyone would disagree that this is a major limitation, but I appreciate the transparancy. Unfortunately the community does not have many viable alternatives at the moment. Transfer performance seem to correlate somewhat with the validation loss.

**Weaknesses:**

I'm overall quite pleased with the paper in its current form. However, there's a couple of thing that I'd like to point out:
- Since the authors are conducting transfer learning experiments in simulation (in addition the real experiments), I would have liked to see more ablations in this setting. Transfer performance seems like a way stronger signal for ablations, e.g., validating that more data does indeed lead to better transfer, not just better validation loss.
- It is not clear to me how the authors arrived at certain experimental setup choices. For example, the authors state on L282 that they use 20-100 demonstrations per task in experiments, but do not specify whether the model is transferred to each task *individually* or *jointly* (i.e., finetuning jointly on N*T demonstrations for T tasks and then evaluating vs. finetuning T models each on N demonstrations). It is also not clear to me which tasks require more demonstrations + whether this relates to the number of tasks considered in a given transfer experiment (e.g. 100 demos per task for Meta-World 5 vs. 20 demos per task for Meta-World 20 would be a reasonable comparison if the model is finetuned jointly on all tasks).

**Questions:**

In addition to addressing my comments above, I would also like the authors to clarify the following:
- L282 states 20-100 demos are used, whereas L300 states that as few as 10 demos are necessary. Can the authors point to such an experiment, if available? Or is this merely an observation made during preparation of the paper? I did not find any such experiments in the appendices.
- Figure 10 (left) seemingly does not include the test loss of the larger model, whereas it is included in Figure 10 (right). Why is that?

**Limitations:**

I believe that limitations are adequately addressed.

---

> ### Author Rebuttal · Authors · 2024-08-04
>
> Thank you so much for the extensive feedback and suggestions. We have revised the manuscript and **conducted new experiments in the attached PDF**  and will address your questions below.
>
> > Since the authors conduct transfer learning experiments in simulation (in addition to real experiments), I would have liked to see more ablations in this setting. Transfer performance seems like a way stronger signal for ablations, e.g., validating that more data does indeed lead to better transfer, not just better validation loss.
>
> In addition to the real-world experiments in **[Table 4]** where we show a better-trained trunk can achieve better performance and compare with several ablations. In **[Figure 16]**, we also show that a better-trained HPT-XL model can outperform an HPT-Base model across all the simulation benchmarks. We also have done additional simulation experiments **[Figure 13b]** on transferring with and without vision and proprioception tokens to illustrate their benefits. Hope this addresses your concerns.
>
> We also copy the general response here for completeness.
>
> We highlight that the experiments in this work are three-fold: pre-training, evaluation in the simulation benchmark (**Figure 16**), and evaluation in the real world (**Figure 14, 15**). Only in the pre-training part, where we need to extensively measure **the scaling behaviors of the learned representations**, do we use the validation losses. It’s not an ideal choice, but it’s a common choice adopted by foundation model training in both LLM and multimodal foundation models. We note that the standard practice of robot learning (specialist model) is to overfit on ~100 trajectories, in which cases validation loss might not be that indicative, and evaluation is not that difficult. But in our case, we are training representations on massive amounts of robot embodiments and trajectories (generalist model), where validation loss should be more representative of the learning progress.
>
> Additionally, we have proposed a simple fix (using smooth L1 loss) to make it more balanced between the difficult and easy parts of the trajectories. This does not include the high costs of evaluating representations in robotics. Previously, validation losses were simply ignored by practitioners, and our work is one step towards that direction. We believe that validation loss can be a continuous metric of representation without running infinite physical experiments at a large scale, and we will further revisit and investigate the correlations at a large scale in future work.
>
> Last but not least, we did many experiments in two sections on transfer learning in simulation and the real world in the submitted version. The simulation experiment is more extensive than previous works. These preliminary experiments have showcased the correlation of validation loss and transfer learning performance (**Table 4 and Figure 16b**)We wanted to highlight that previous works pre-trained on Open-X have evaluated mostly using the training embodiments, which we do not have access to. Our real-world evaluation tasks are also more challenging than previous works.
>
> > For example, the authors state on L282 that they use 20-100 demonstrations per task in experiments, but do not specify whether the model is transferred to each task individually or jointly (i.e., finetuning jointly on N*T demonstrations for T tasks and then evaluating vs. finetuning T models each on N demonstrations).
>
> Other than the meta-world setup (where single-task evaluation becomes too easy), all experiments, including the real-world task, have training conducted on each task individually, as a standard setup. So we use “finetuning T models each on N demonstrations setup.”
>
> > It is also not clear to me which tasks require more demonstrations + whether this relates to the number of tasks considered in a given transfer experiment (e.g. 100 demos per task for Meta-World 5 vs. 20 demos per task for Meta-World 20 would be a reasonable comparison if the model is finetuned jointly on all tasks).
>
> The only joint multitask experiment is meta-world. In that case, we try to use a demonstration number such that the trained policies performance is within a reasonable range, between 50 and 90 percent success rates. indeed, when we train with more tasks, we do not have to use the same number of trajectories for each task. There is no particular rule for which tasks require demonstrations in practice. It mostly depends on the learner's performance.
>
> > L300 states that as few as 10 demos are necessary. Can the authors point to such an experiment, if available? Or is this merely an observation made during preparation of the paper?
>
> The 10 demo requirement is an observation that we made on the robomimic Lift task, which is arguably a simple simulation benchmark task.
>
> > Figure 10 (left) seemingly does not include the test loss of the larger model, whereas it is included in Figure 10 (right). Why is that?
>
> For scaling ablation experiments, we mostly use the pre-training experiments as well as the task performance success rates as the metric, as the success rate has been brought up to be more important for robotics. We have completed a new set of experiments with different scales of pre-trained models. Please see **Figure 16(b)** for more details.
>
>
> Thank you very much again for your constructive feedback. Please do not hesitate to let us know if you have any other questions and/or comments.

---

> > ### Comment · Reviewer_AQbN · 2024-08-09
> >
> > Thank you for responding to my questions in such detail. I have read the author rebuttal, responses to my questions, as well as those of my fellow reviewers. I believe that my concerns have been addressed and I have updated my score from 6 -> 7 to reflect that. This score is contingent on the authors incorporating reviewer feedback into the camera-ready version of their paper, especially the clarifications on experimental setup (finetuning T models each on N demonstrations), as well as the new experiments that were conducted during the rebuttal.

---

### Official Review · Reviewer_Vi4L · 2024-07-16

**Soundness:** 3
**Presentation:** 2
**Contribution:** 2
**Rating:** 6
**Confidence:** 4

**Summary:**

This paper presents an approach for training large transformer behavior models on diverse heterogenous data. This involves separate encoders for proprioception and image observations for each embodiment, as well as different action decoding heads. Experiments include validation loss comparisons.

**Strengths:**

While training large models that can learn from diverse heterogenous datasets is an important problem of relevance to the community, the current paper has some key limitations, please see weaknesses.

**Weaknesses:**

After rebuttal - Concerns regarding validation loss vs success rate has been addressed with additional experiments, claims have been further strengthened with real world results

########################

1. Evaluation Metrics in the experiments

For training robot behavior models, the standard metric used in the community is the success rate of the trained model after being fine-tuned on a few demos for a new task [1,2,3]. The success rate measures the robot's ability to actually perform tasks of interest, using the large pre-trained weights. This is a more accurate metric than validation loss for how to judge if the large scale model training is making use of the diverse quantities of training data. Validation loss is not perfectly correlated with downstream robot performance. If it is difficult/cumbersome to do these in the real world, this analysis should atleast be done in simulation. The reason validation loss is a much weaker indicator in the robotics setting is because it doesn't measure how good the model is on the on-policy distribution, which can diverge wildly from the expert data distribution used for training.

2. Design choice justification

The proposed approach has separate encoders and decoders for each observation and action modality. This seems like it could lead to less sharing of information across diverse datasets, leading to poorer transfer capabilities. Why should we not instead have a single action decoding head, as is done in [2]? What if the same image embedding backbone is used for all all observations - this would lead to the sharing of features potentially enabling better transfer. This choice of separate encoders/decoders for each modality is not sufficiently ablated and analyzed.







[1]: RT1-Robotics Transformer for Real-World Control at Scale, Brohan et al.

[2]: Octo - Octo: An Open-Source Generalist Robot Policy, Octo Model Team

[3]: OpenVLA: An Open-Source Vision-Language-Action Model, Kim et al.

**Questions:**

1. How does the presented approach perform when deployed on real/simulated robots in terms of success rate, and how does it compare to previous approaches [1,2,3] ? Is it important to have separate encoders/decoder for each observation/action modality?








[1]: RT1-Robotics Transformer for Real-World Control at Scale, Brohan et al.

[2]: Octo - Octo: An Open-Source Generalist Robot Policy, Octo Model Team

[3]: OpenVLA: An Open-Source Vision-Language-Action Model, Kim et al.

---

> ### Author Rebuttal · Authors · 2024-08-04
>
> Thank you so much for the extensive feedback and suggestions. We have revised the manuscript and **conducted new experiments in the attached PDF** and will address your questions below.
>
> > Evaluation Metrics in the experiments. Validation loss is not perfectly correlated with downstream robot performance. If it is difficult/cumbersome to do these in the real world, this analysis should at least be done in simulation.
>
> Thanks for bringing this up and we are aware of the limitation on validation loss. Therefore, significant experiments have been done in simulation (**Figure 16 b**). We have shown that the more well-trained model HPT-XL can outperform HPT-base in the simulation benchmark. Moreover, we have added additional experiments to compare against several generalist policies in the recently released Simpler Benchmark. See **Figure 13(a)** for the result. While evaluation and metrics are still huge issues for robotics and generalist models, we believe that our simulation experiments are more extensive than previous works such as RT, Octo, and OpenVLA. Moreover, we kindly note that OpenVLA is released **after** the paper is submitted.
>
> We also copy the general response here for completeness.
>
> - We highlight that the experiments in this work are three-fold: pre-training, evaluation in the simulation benchmark (**Figure 16**), and evaluation in the real world (**Figure 14, 15**). Only in the pre-training part, where we need to extensively measure **the scaling behaviors of the learned representations**, do we use the validation losses. It’s not an ideal choice, but it’s a common choice adopted by foundation model training in both LLM and multimodal foundation models. As an analogy, there are also potential issues raised on the validity of perplexity for a small dataset in language modeling (which can also have out-of-distribution issues), but it is generally acceptable that perplexity is a good metric for studying large-scale pre-training these days.
>
> - We note that the standard practice of robot learning (specialist model) is to overfit on ~100 trajectories with one day of training, in which cases validation loss might not be that indicative, and evaluation is not that difficult. But in our case, we are training with hundreds of GPUs and weeks of time with massive amounts of robot embodiments and trajectories (generalist model), where validation loss should be more representative of the learning progress.
>
> - Additionally, we have proposed a simple fix (using smooth L1 loss) to make it more balanced between the difficult and easy parts of the trajectories. This does not include the high costs of evaluating representations in robotics. Previously, validation losses were simply ignored by practitioners, and our work is one step towards that direction. We believe that validation loss can be a more continuous reflection of representation at a large scale, and we will further revisit and investigate the correlations at a large scale in future work.
>
> - Last but not least, we did many experiments in section 5 on transfer learning in simulation and the real world in the submitted version. The simulation experiment is more extensive than previous works with thousands of runs. These experiments have showcased the correlation between validation loss and transfer learning performance (**Table 4 and Figure 16b**), where larger and more well-trained models indeed can perform better. We wanted to highlight that previous works pre-trained on Open-X have been evaluated mostly using the training embodiments, which we do not have access to and we argue that measuring representation learning performance in a novel environment is also a valid choice. Our real-world evaluation tasks are also more challenging than previous works.
>
>
>
> > The proposed approach has separate encoders and decoders for each observation and action modality. This seems like it could lead to less sharing of information across diverse datasets, leading to poorer transfer capabilities. Why should we not instead have a single action decoding head, as is done in Octo?
>
> The key motivation is that different observation and action spaces are very heterogeneous: they have different dimensions and live in different spaces. We want to separate out the part that can be potentially shared, which is the middle part of the network that contains more abstract representations. The proposed method is one architecture choice to handle such issues.
>
> **Encoder:** It is technically impractical to map different dimensions of tensors (in proprioception) to the same space with the same network, unless we have a unified tokenization method. Our proposed cross-attention (or resampler) is one method to unify different inputs. In Octo, they do not consider proprioception and therefore can use a domain-specific token to represent the heterogeneity. When they need to handle the proprioception, they still need to reinitialize a new encoder for such domain and input (see their Berkeley Insertion experiment).
>
>
> **Decoder**: We want to highlight that each of the action spaces can be drastically different from one another in robotics (such as pose and joint angles). Moreover, a finetuning procedure is still needed for any generalist model that we discuss in robotics. Octo trains a shared decoder based on transformer architecture and needs to reinitialize the task token during finetuning. In some cases, they still have to finetune the decoder head. The compute efficiency and the generalist design are not that different from our case. Indeed, in the early part of the project, we also experimented with Octo-like architecture with task-specific decoder tokens, but we did not find a big difference. We prefer to not expose the task information to the trunk to ensure it can learn task-agnostic representations.
>
>
> (continued)

---

> ### Author Response · Authors · 2024-08-04
> **Response to Review of Submission7169 by Reviewer Vi4L (Continued)**
>
> We also copy the general response here for completeness.
>
> - We highlight that the main motivation to tokenize embodiments and align them in the same latent space, is due to the practical constraints of heterogeneity in **proprioception** and action spaces. These input and output tensors with different dimensions cannot be handled with a **single network** technically (think about two weight matrices with different dimensions), and are **not considered** in previous works before. To handle these, we use a lightweight encoder-decoder approach in this project. The idea is partly motivated by the “experts” in the recent mixture of expert literature as well as the “projector” or “connector” in the multimodal literature. For the vision modalities, we actually **share** the vision encoder and only use a small adaptor to project the frozen vision encoders.
>
> - Therefore, this is also a meta-level choice that is brought by heterogeneous pre-training, and is independent of specific architecture choices (shared head or separate heads). While this HPT architecture is a specific architectural choice that we made, the idea is motivated to handle **more heterogeneity across domains and sensor modalities**. This is a more **significant point and is different** than previous works. There are more complex methods such as using GNN or a unified tokenizer across these embodiments to handle this issue, but we leave those for future work.
>
> - Moreover, this simple and general architecture choice is **not** conflicted with sharing information across embodiments because the encoders (stem) and decoders (head) do not contain many parameters, and are merely applied to match the dimensions and project tensors in different spaces to the shared latent space, similar to how multimodal foundation models are trained. Moreover, in all robotic generalist works including Octo and OpenVLA, finetuning is still necessary,  which means some information in the new embodiment is not learned during pre-training.
>
> > What if the same image embedding backbone is used for all observations - this would lead to the sharing of features potentially enabling better transfer.
>
> Thanks for bringing this up. Since we are not finetuning the pre-trained frozen vision encoders, all the image embedding backbones are actually shared across observations from different embodiments. Only the adaptation layers are different for embodiments, which are meant to project the vision features to the shared latent space, as commonly used in the multimodal learning literature. In this work, we only explore single-stage pre-training, finetuning, or joint training of vision encoders can potentially happen in the later stages of pre-training. We will leave those for future work.
>
>
> > How does the presented approach perform when deployed on real/simulated robots in terms of success rate, and how does it compare to previous approaches [1,2,3]?
>
> We have added additional experiments to compare against several generalist policies in the recently released Simpler simulation benchmark. See **[Figure 13a]** for the result. Specifically, we compare the models across 3 different tasks in the Google GDR embodiment with the visual matching setup. We have found HPT to perform quite well on these benchmarks compared to Octo for instance. Also, OpenVLA is released after our submission and RT1 does not support finetuning. Hope this addresses your concern.
>
> > Is it important to have separate encoders/decoder for each observation/action modality?
>
> Please see our earlier answers. As for the benefit or motivation for creating separate encoders/decoders, we are targeting a simplistic approach to handling heterogeneous data and maximizing the information that can be shared. Critically, the separate encoders/decoders are relatively small (2% of the parameters) compared to the trunk. The majority of the information is contained in the pre-trained encoders as well as the shared trunk.
>
>
> Thank you very much again for your constructive feedback. Please do not hesitate to let us know if you have any other questions and/or comments.

---

> > ### Author Response · Authors · 2024-08-09
> > **Response to Review of Submission7169 by Reviewer Vi4L (Continued)**
> >
> > Dear Reviewer, we would like to thank you again for your efforts and time in providing thoughtful feedback and comments. We’ve conducted additional experiments according to your suggestions and replied to all the questions and concerns. Since the discussion period is ending soon, we would greatly appreciate it if you could let us know whether you have any additional comments. Thanks a lot!

---

> > > ### Comment · Reviewer_AQbN · 2024-08-09
> > >
> > > > For training robot behavior models, the standard metric used in the community is the success rate of the trained model after being fine-tuned on a few demos for a new task [1,2,3]. The success rate measures the robot's ability to actually perform tasks of interest, using the large pre-trained weights. This is a more accurate metric than validation loss for how to judge if the large scale model training is making use of the diverse quantities of training data. Validation loss is not perfectly correlated with downstream robot performance. If it is difficult/cumbersome to do these in the real world, this analysis should atleast be done in simulation.
> > >
> > > > How does the presented approach perform when deployed on real/simulated robots in terms of success rate, and how does it compare to previous approaches [1,2,3] ?
> > >
> > > The authors conduct transfer learning experiments in simulation + real and report success rates. They don't compare to OpenVLA [3] since it was not released at the time of submission. I don't think these comments are appropriate at all.

---

> > > > ### Author Response · Authors · 2024-08-11
> > > > **Thank you, Reviewer AQbN!**
> > > >
> > > > We would like to thank reviewer AQbN for the response and for sharing your opinions in this review!

---

> > > > > ### Comment · Reviewer_Vi4L · 2024-08-13
> > > > > **Response to Author Reply**
> > > > >
> > > > > I thank the authors for their detailed response and the extra experiments conducted. I apologize for asking a comparison to OpenVLA [3] since it was released after the submission, this was an oversight. I had originally mentioned it in the weaknesses section of the review to point to methods that use success rate instead of validation loss to compare performance. This was to substantiate the point that validation loss is not typically used in the community to compare methods, and that the efficacy of the approach would need success rate on tasks instead. With the additional experiments in Fig 13.a as well as Fig 16.b, these concerns have been addressed. I also appreciate the extra real world experiments with a new robot embodiment.
> > > > >
> > > > > Given this, I am increasing my score (3->6)

---

> > > > > > ### Author Response · Authors · 2024-08-13
> > > > > > **Thank you, Reviewer Vi4L!**
> > > > > >
> > > > > > Thank you so much for the response, Reviewer Vi4L!

---

### Author Rebuttal · Authors · 2024-08-04

We thank all the reviewers for their valuable and constructive comments. Overall, we appreciate that most reviewers acknowledged the contributions, **“important problem of relevance to the community.”** (from Reviewer Vi4L), **“The proposed architecture is fairly simple and intuitive.”** (from Reviewer AQbN) and noted the paper shows **“pre-training policies across heterogeneity can exhibit scaling behaviors of validation losses.”** (from Reviewer wUp5) and is **“Insightful scaling analysis, which does not exist in the context of robotics yet”** (from Reviewer wUp5).

In the two months since the submission, we have worked continuously on the paper to improve the quality. Many questions and concerns raised by the reviewers are answered through our additional experiments and we hope that reviewers  take the new experiments into consideration when making their final recommendations. We have attached a **rebuttal PDF summary** of Tables and Figures for the **additional experiments** (with highlights in blue) and will discuss them below.

**Additional Experiments:**

**Simulation**: We redesigned **[Figure 16 (b)]** and added policy fine-tuning performances from larger HPT backbones. We conducted additional simulation experiments such as ablating the effects of vision and proprioception tokens in **[Figure 13 (b)]**. In **[Figure 13(a)]**, we discussed new experiments on the **Simpler [2] benchmark which demonstrated strong performance compared with other generalist models including RT-1, RT-2, and Octo.** In **Figure 17**, we highlight our extensive evaluation of the simulation across 4 different benchmarks and thousands of runs. [Reviewer Vi4L,wUp5,KFGf,AQbN]. Additionally, we have included a novel deployed robot dataset Frodobot [1]. We have revised the experiment in **[Figure 16 (a)]** and focused more on the effects of adding embodiment datasets to the pre-training performance.


 **Real-world**: We have added more sweep-food experiments with larger pre-trained trunks in **[Table 4]**. We ablate against pre-trained models without proprioception and compare with more baselines. Importantly, we also added additional experiments on the insertion task with a **new robot embodiment** with different observation and action spaces, shown in **Figure 14**. These new experiments further validate the performance of HPT pre-trained model. We have not been able to make Octo finetuning work well in our real-world setup despite weeks of effort, so its current finetuning performance does not match really well. OpenVLA is **released after our submission**, and its inference GPU requirement exceeds our real robot setup. [KFGf]



[1] “FrodoBots-2K Dataset”, FrodoBots Lab, 2024

[2] “Evaluating real-world robot manipulation policies in simulation”, Li et al., 2024.


Now we will summarize a few answers that hopefully address some of the common questions posed by reviewers and we will also briefly clarify a few points.

***Key Answers***



**1. Validation loss and evaluation metric [Vi4L,AQbN,wUp5]**:

- We highlight that the experiments in this work are three-fold: pre-training, evaluation in the simulation benchmark (**Figure 16**), and evaluation in the real world (**Figure 14, 15**). Only in the pre-training part, where we need to extensively measure **the scaling behaviors of the learned representations**, do we use the validation losses. It’s not an ideal choice, but it’s a common choice adopted by foundation model training in both LLM and multimodal foundation models. As an analogy, there are also potential issues raised on the validity of perplexity for a small dataset in language modeling (which can also have out-of-distribution issues), but it is generally acceptable that perplexity is a good metric for studying large-scale pre-training these days.

- We note that the standard practice of robot learning (specialist model) is to overfit on ~100 trajectories with one day of training, in which cases validation loss might not be that indicative, and evaluation is not that difficult. But in our case, we are training with hundreds of GPUs and weeks of time with massive amounts of robot embodiments and trajectories (generalist model), where validation loss should be more representative of the learning progress.

- Additionally, we have proposed a simple fix (using smooth L1 loss) to make it more balanced between the difficult and easy parts of the trajectories. This does not include the high costs of evaluating representations in robotics. Previously, validation losses were simply ignored by practitioners, and our work is one step towards that direction. We believe that validation loss can be a more continuous reflection of representation at a large scale, and we will further revisit and investigate the correlations at a large scale in future work.

- Last but not least, we did many experiments in section 5 on transfer learning in simulation and the real world in the submitted version. The simulation experiment is more extensive than previous works with thousands of runs. These experiments have showcased the correlation between validation loss and transfer learning performance (**Table 4 and Figure 16b**), where larger and more well-trained models indeed can perform better. We wanted to highlight that previous works pre-trained on Open-X have been evaluated mostly using the training embodiments, which we do not have access to and we argue that measuring representation learning performance in a novel environment is also a valid choice. Our real-world evaluation tasks are also more challenging than previous works.

(continued)

---

### Author Response · Authors · 2024-08-04
**General Response to All Reviewers (Continued)**

**2. Lack comparison with other generalist models [Vi4L,wUp5,KFGf]**: In **[Figure 13(a)]**, we discussed new experiments on the **Simpler [2] benchmark which demonstrated strong performance compared with other generalist models including RT-1, RT-2, and Octo.** This simulation experiment shows that our model is comparable to state-of-the-art models. We note that it can be challenging to apply the zero-shot version of these models to our setting and we also do not have access to their hardware. We also note that works like OpenVLA, brought by reviewer Vi4L, are released after our submission.

**3. Architecture Designs [Vi4L,wUp5]**: We want to provide some clarifications on our architecture design and hopefully shed light on our motivation.

- We highlight that the main motivation to tokenize embodiments and align them in the same latent space, is due to the practical constraints of heterogeneity in **proprioception and action spaces**. Without any pre/post-processing, these input and output tensors with different dimensions cannot be handled with a **single network** (think about two weight matrices with different dimensions), and are **not considered** in previous works before. To handle these, we use a simple and efficient encoder-decoder approach in this project. The idea is partly motivated by the “experts” in the recent mixture of expert literature as well as the “projector” or “connector” in the multimodal literature. For the vision modalities, we actually **share** the vision encoder and only use a small adaptor to project the frozen vision encoders.

- This is also a meta-level choice that is brought by heterogeneous pre-training, and is independent of specific architecture choices (shared head or separate heads). While this HPT architecture is a specific architectural choice that we made, the idea is motivated to handle **more heterogeneity across domains and sensor modalities**. This is a more **significant point and is different** than previous works. There are more complex methods such as embedding embodiment information, using paddings, GNN or a unified tokenizer across these embodiments to handle this issue, but we leave those for future work.

- Moreover, this simple and general architecture choice is **not** conflicted with sharing information across embodiments because the encoders (stem) and decoders (head) do not contain many parameters, and are used as simple methods applied to match the dimensions and project tensors in different spaces to the shared latent space, similar to how multimodal foundation models are trained. Moreover, similar to other generalist policy works including Octo and OpenVLA, finetuning is used at test time to adapt well to new embodiment and tasks.

**4. More real-world experiments and ablation [KFGf]**: We also added additional experiments on the insertion task with a **different robot embodiment**, shown in **[Figure 14]**. This task uses a different observation and action space and requires more precision than the pet-care setting. We have also made ablation studies on different pre-training models in the real world in Table 4. These new experiments further validate the observation on HPT pre-trained model.


We thank the reviewers again for their raised points. I hope our new experiments and clarifications addressed some of the comments and concerns of the reviewers.

---

### Decision · Program_Chairs · 2024-09-25

**Decision:**

Accept (spotlight)

**Comment:**

This submission proposed a new method, called HPT, for quick (task, embodiment, and environment) adaptation by pretraining a policy representation from large datasets. In order to encourage shared learning across multi-modal data, the model divides the policy into three parts: a dataset-specific token encoder (Stem), a shared processing trunk, and dataset-specific action heads. The authors showed that this approach of dividing into three parts helps effective scaling and adaptability on both simulated and real-world tasks. The results are shown by finetuning with few demonstration in the Open-X dataset.

Overall, reviewers agreed that the method and the results are compelling and it should be shared with the community. There are several experiments and analysis reviewers asked for to further improve the submission. Please add them to your camera-ready copy.